# Mix-of-Show: Decentralized Low-Rank Adaptation for Multi-Concept Customization of Diffusion Models

**Yuchao Gu**[1], **Xintao Wang**[3], **Jay Zhangjie Wu**[1], **Yujun Shi**[2], **Yunpeng Chen**[2],
**Zihan Fan**[2], **Wuyou Xiao**[2], **Rui Zhao**[1], **Shuning Chang**[1], **Weijia Wu**[1],
**Yixiao Ge**[3], **Ying Shan**[3], **Mike Zheng Shou**[1]*

[1]Show Lab, [2]National University of Singapore    [3]ARC Lab, Tencent PCG

https://showlab.github.io/Mix-of-Show

## Abstract

Public large-scale text-to-image diffusion models, such as Stable Diffusion, have gained significant attention from the community. These models can be easily customized for new concepts using low-rank adaptations (LoRAs). However, the utilization of multiple concept LoRAs to jointly support multiple customized concepts presents a challenge. We refer to this scenario as decentralized multi-concept customization, which involves single-client concept tuning and center-node concept fusion. In this paper, we propose a new framework called Mix-of-Show that addresses the challenges of decentralized multi-concept customization, including concept conflicts resulting from existing single-client LoRA tuning and identity loss during model fusion. Mix-of-Show adopts an embedding-decomposed LoRA (ED-LoRA) for single-client tuning and gradient fusion for the center node to preserve the in-domain essence of single concepts and support theoretically limitless concept fusion. Additionally, we introduce regionally controllable sampling, which extends spatially controllable sampling (*e.g.*, ControlNet and T2I-Adapter) to address attribute binding and missing object problems in multi-concept sampling. Extensive experiments demonstrate that Mix-of-Show is capable of composing multiple customized concepts with high fidelity, including characters, objects, and scenes.

## 1  Introduction

Open-source text-to-image diffusion models, such as Stable Diffusion [1], empower community users to create customized models by collecting personalized concept images and fine-tuning them with low-rank adaptation (LoRA) [2, 3]. These tailored LoRA models achieve unparalleled quality for specific concepts through meticulous data selection, pre-processing, and hyperparameter tuning. While existing concept LoRAs serve as plug-and-play plugins for pretrained models, there are still challenges in utilizing multiple concept LoRAs to extend the pretrained model and enable joint composition of those concepts. We refer to this

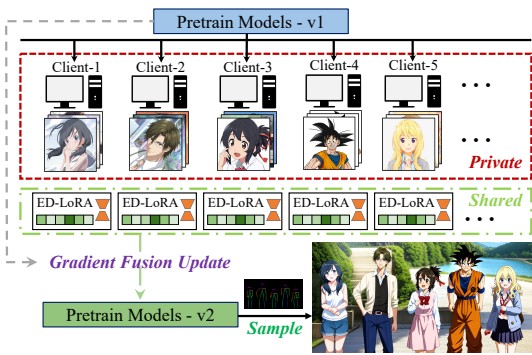

Figure 1: Illustration of decentralized multi-concept customization via Mix-of-Show.

---

*Corresponding Author.

37th Conference on Neural Information Processing Systems (NeurIPS 2023).

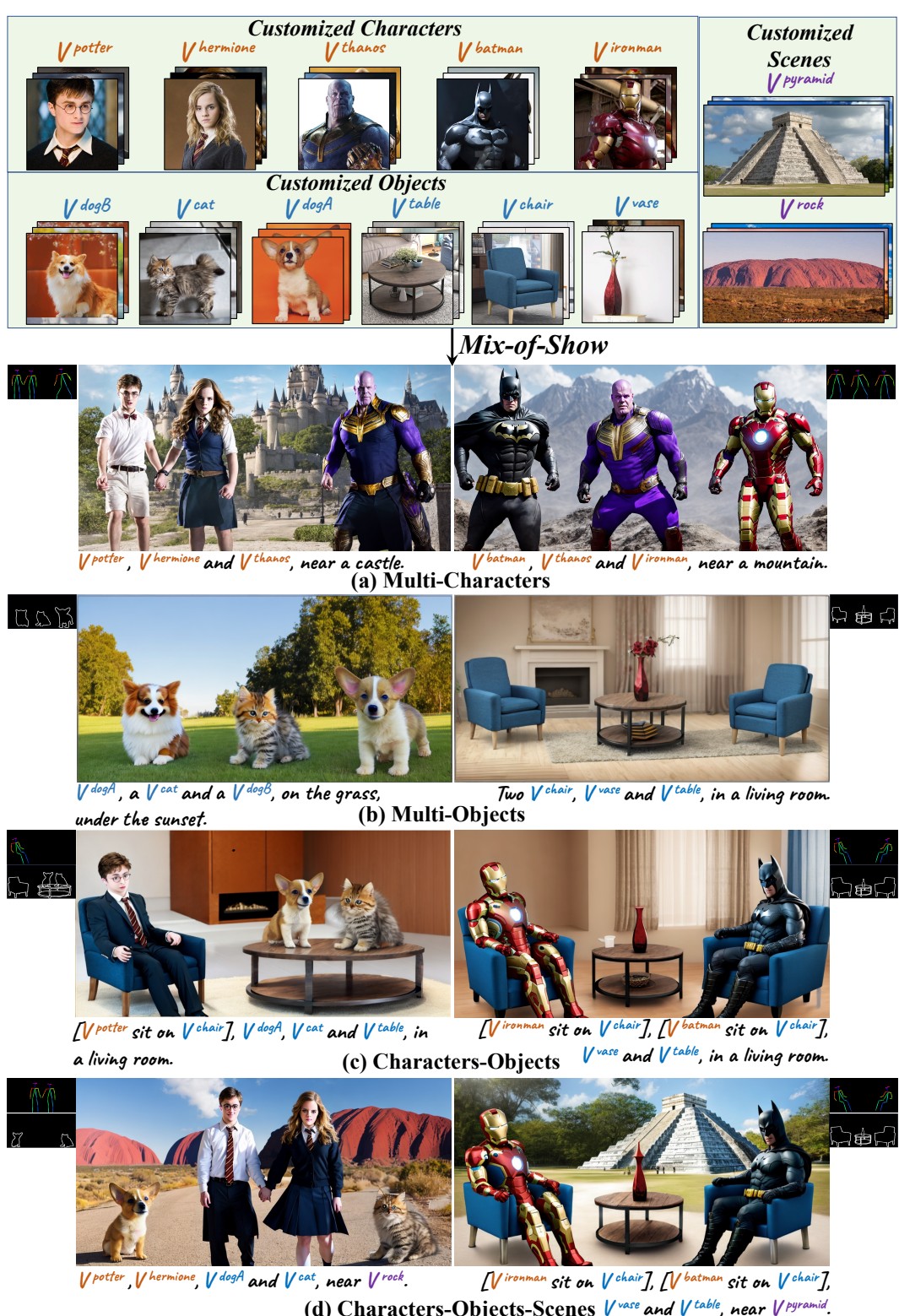

Figure 2: How to generate Harry Potter and Thanos, these two (or even more) concepts from different shows, in the same image? Our Mix-of-Show enables complex compositions of multiple customized concepts (*e.g.*, characters, objects, scenes) with individually trained concept LoRAs.

scenario as decentralized multi-concept customization. As shown in Fig. 1, it involves two steps: single-client concept tuning and center-node concept fusion. Each client retains their private concept data while sharing the tuned LoRA models. The center node leverages these concept LoRAs to update the pretrained model, enabling joint sampling of these customized concepts. Decentralized multi-concept customization facilitates maximum community engagement in producing high-quality concept LoRAs and offers flexibility in reusing and combining different concept LoRAs.

However, the existing LoRA tuning and weight fusion techniques [3] fail to address the challenges of decentralized multi-concept customization. We have identified two main challenges: concept conflict and identity loss. Concept conflict arises because current LoRA tuning methods do not differentiate between the roles of embeddings and LoRA weights. Our research reveals that embeddings effectively capture concepts within the pretrained models' domain, while LoRA weight assist in capturing out-of-domain information (*e.g.*, styles or fine details cannot be directly modeled by the pretrained model). However, existing LoRA tuning methods place excessive emphasis on LoRA weights while overlooking the importance of embeddings. Consequently, the LoRA weights encode a significant portion of the identity of a given concept, resulting in semantically similar embeddings being projected onto concepts with different appearances. This, in turn, leads to conflicts during model fusion. Furthermore, existing weight fusion strategies compromise each concept's identity and introduce interference from other concepts by performing a weighted average of all concept LoRAs.

To overcome the challenges of decentralized multi-concept customization, we propose Mix-of-Show, which involves embedding-decomposed LoRA (ED-LoRA) for single-client tuning and gradient fusion for center-node fusion. In single-client tuning, ED-LoRA is designed to address concept conflicts by preserving more in-domain essence within the embedding. To achieve this, we enhance the expressive ability of the concept embedding by decomposing it into layer-wise embeddings [4] and multi-word representations. At the central node, gradient fusion leverages multiple concept LoRAs to update the pretrained model. Since the diffusion model includes both forward and reverse diffusion processes, we can obtain the input/output features of each layer through sampling, even in the absence of data. Features from multiple concept LoRAs are combined to generate the fused gradient, which is used for layer-wise updating. Compared to weight fusion [3], gradient fusion aligns the inference behavior of each individual concept, significantly reducing identity loss.

To demonstrate the capabilities of Mix-of-Show, we introduce regionally controllable sampling for multi-concept generation. Direct multi-concept generation often encounters issues such as missing objects and attribute binding [5, 6]. Recently, spatially controllable sampling (*e.g.*, ControlNet [7], T2I-Adapter [8]) have been introduced to guide diffusion models using spatial hints (*e.g.*, keypose or sketch), which resolve the problem of missing objects but still faces challenges of attribute binding in multi-concept generation. Considering that spatial layout is pre-defined when adopting spatial conditions, we propose injecting region prompts through regional-aware cross-attention. Powered by Mix-of-Show and regionally controllable sampling, we can achieve complex compositions of multiple customized concepts, including characters, objects, and scenes, as illustrated in Fig. 2. In summary, our contributions are as follows: 1) We analyze the challenges of decentralized multi-concept customization. 2) We propose the Mix-of-Show framework, consisting of an embedding-decomposed LoRA (ED-LoRA) and gradient fusion, to address the concept conflict and identity loss in decentralized multi-concept customization. 3) We introduce regionally controllable sampling to demonstrate the potential of Mix-of-Show in composing multiple customized concepts.

## 2 Related Work

### 2.1 Concept Customization

Concept customization aims to extend pretrained diffusion models to support personalized concepts using only a few images. There are two main types of concept tuning methods: embedding tuning (*e.g.*, Textual Inversion [9] and P+ [4]) and joint embedding-weight tuning (*e.g.*, Dreambooth [10] and Custom Diffusion [11]). Additionally, the community [3] adopts low-rank adapter (LoRA) [2] for concept tuning, which is lightweight and can achieve comparable fidelity to full weight tuning.

Although significant progress has been made in single-concept customization, multi-concept customization remains a challenge. Custom Diffusion [11] proposes co-training of multiple concepts or constrained optimization of several existing concept models. Following this, SVDiff [12] introduces data augmentation to prevent concept mixing in co-training multi-concepts, and Cones [13] discovers

concept neurons that can be added to support multiple concepts. However, their methods are typically restricted to fuse 2-3 semantically distinct concepts. In contrast, Mix-of-Show can combine theoretically limitless customized concepts, including those within the same semantic category.

Another research line in concept customization, as explored in studies by Instantbooth [14], ELITE [15], and Jia et al. [16], focuses on achieving fast test-time customization. These methods involve pretraining an encoder on a large-scale dataset specific to the desired category. During inference, when provided with a few representative concept images from the trained category, the encoder extracts features that complement the pretrained diffusion models and support customized generation. However, these methods require training a separate encoder for each category, typically limited to common categories (*e.g.*, person or cats). This limitation hinders their ability to customize and compose more diverse and open-world subjects.

### 2.2 Decentralized Learning

Decentralized or federated learning aims to train models collaboratively across different clients without sharing data. The *de facto* algorithm for federated learning, FedAvg, was proposed by [17]. This method simply averages the weights of each client's model to obtain the final model. However, we find that directly applying this simple weight averaging is not ideal for fusing LoRAs of different concepts. To improve over FedAvg, previous works have either focused on local client training [18, 19, 20, 21, 22, 23, 24] or global server aggregation [25, 26, 27, 28, 29, 30]. Motivated by this, we explore the optimal design of single-client tuning and center-node fusion for decentralized multi-concept customization.

### 2.3 Controllable Multi-Concept Generation

Direct multi-concept generation using text prompts alone faces challenges such as missing objects and attribute binding [6, 31, 32, 33, 34]. Previous approaches, like Attend-and-Excite [5] and Structure Diffusion [6], have attempted to address these issues, but the problem still persist, limiting the effectiveness of multi-concept generation. Recent works, such as ControlNet [7] and T2I-Adapter [8], introduce spatial control (*e.g.*, keypose and sketch) and enable more accurate compositions, resolving the problem of missing objects in multi-concept generation. However, attribute binding remains a challenge. In our work, we tackle this challenge through regionally controllable sampling.

## 3 Methods

In this section, we provide a brief background on text-to-image diffusion models and concept customization in Sec. 3.1. We then introduce the task formulation of decentralized multi-concept customization in Sec. 3.2, followed by a detailed description of our method in Sec. 3.3 and Sec. 3.4.

### 3.1 Preliminary

**Text-to-Image Diffusion Models.** Diffusion models [35, 36, 37, 38, 39, 40, 41] belong to a class of generative models that gradually introduce noise into an image during the forward diffusion process and learn to reverse this process to synthesize images. When combined with pretrained text embeddings, text-to-image diffusion models [1, 42, 43, 44, 45, 46] are capable of generating high-fidelity images based on text prompts. In this paper, we conduct experiments using Stable Diffusion [1], which is a variant of the text-to-image diffusion model operating in the latent space. Given a condition $c = \psi(P^*)$, where $P^*$ is the text prompt and $\psi$ is the pretrained CLIP text encoder [47], the training objective for stable diffusion is to minimize the denoising objective by

$$\mathcal{L} = \mathbb{E}_{z,c,\epsilon,t}[\|\epsilon - \epsilon_\theta(z_t, t, c)\|_2^2], \tag{1}$$

where $z_t$ is the latent feature at timestep $t$ and $\epsilon_\theta$ is the denoising unet with learnable parameter $\theta$.

**Embedding Tuning for Concept Customization.** Textual Inversion [9] represents the input concept using a unique token $V$. When provided with a few images of the target concept, the embedding of $V$ is tuned using Eq. 1. After tuning, the embedding for $V$ encodes the essence of the target concept and functions like any other text in the pretrained model. To achieve greater disentanglement and control, P+ [4] introduces layer-wise embeddings for concept tokens, denoted as $V^+$ in this paper.

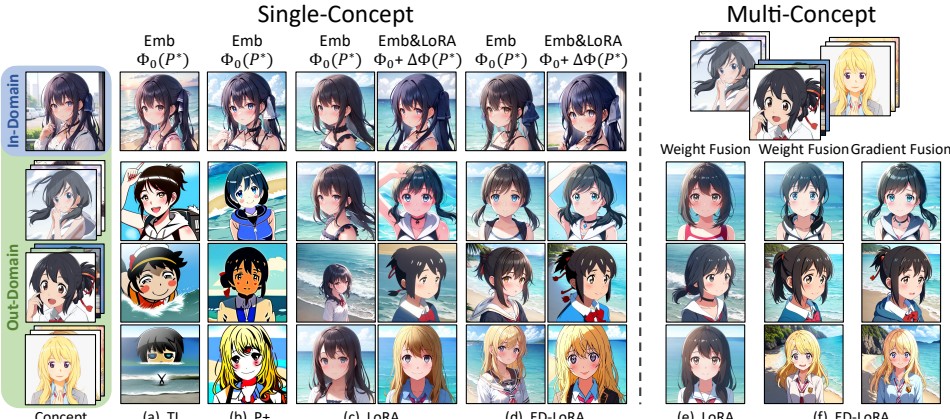

Figure 3: Single- and multi-concept customization between the embedding tuning (*i.e.*, Textual Inversion (TI) [9] and P+ [4]), and joint embedding-weight tuning (*i.e.*, LoRA [3] and our ED-LoRA). $P^*$ = "Photo of a $V$, near the beach". $\Phi_0$ and $\Delta\Phi$ denotes the pretrained model and LoRA weight.

**Low-Rank Adaptation.** Low-rank adaptation (LoRA) [2] was initially proposed to adapt large-language models to downstream tasks. It operates under the assumption that weight changes during adaptation have a low "intrinsic rank" and introduces a low-rank factorization of the weight change to obtain the updated weight $W$, which is given by $W = W_0 + \Delta W = W_0 + BA$. Here, $W_0 \in \mathbb{R}^{d \times k}$ represents the original weight in the pretrained model, and $B \in \mathbb{R}^{d \times r}$ and $A \in \mathbb{R}^{r \times k}$ represent the low-rank factors, with $r \ll \min(d, k)$. Recently, the community [3] has adopted LoRA for fine-tuning diffusion models, leading to promising results. LoRA is typically used as a plug-and-play plugin in pretrained models, but the community also employs weight fusion techniques to combine multiple LoRAs:

$$W = W_0 + \sum_{i=1}^{n} w_i \Delta W_i, \quad \text{s.t.} \sum_{i=1}^{n} w_i = 1, \tag{2}$$

where $w_i$ denotes the normalized importance of different LoRAs.

## 3.2 Task Formulation: Decentralized Multi-Concept Customization

While custom diffusion [11] has attempted to merge two tuned concepts models into a pretrained model, their findings suggest that co-training with multiple concepts yields better results. However, considering scalability and reusability, we focus on merging single-concept models to support multi-concept customization. We refer to this setting as decentralized multi-concept customization.

Formally, decentralized multi-concept customization involves a two-step process: single-client concept tuning and center-node concept fusion. As shown in Fig. 1, each of the $n$ clients possesses its own private concept data and tunes the concept model $\Delta W_i$. Here, $\Delta W_i$ represents the changes in network weights, which specifically refers to LoRA weights in our work. We omit discussing the merging of text embeddings, as the tuned embeddings can be seamlessly integrated into the pretrained model without conflicts.

After tuning, the center node gathers all LoRAs to obtain the updated pretrained weight $W$ by:
$$W = f(W_0, \Delta W_1, \Delta W_2, \ldots, \Delta W_n), \tag{3}$$
where $f$ represents the update rule that operates on the original pretrained model weight $W_0$ and the $n$ concept LoRAs $\{\Delta W_i, i = 1 \cdots n\}$. One straightforward updating rule $f$ is weight fusion, as illustrated in Eq. 2. Once updated, the new model $W$ should be capable of generating all the concepts introduced in the $n$ LoRAs.

## 3.3 Mix-of-Show

In this section, we introduce Mix-of-Show, containing ED-LoRA (in Sec. 3.3.1) for single-client concept tuning, gradient fusion (in Sec. 3.3.2) for center-node concept fusion.

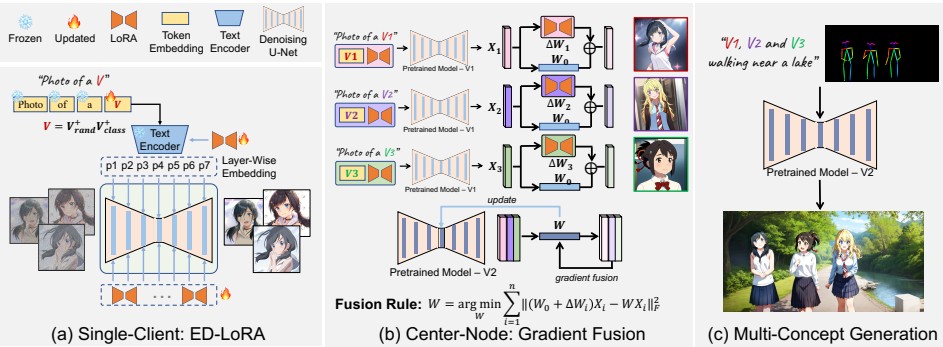

Figure 4: Pipeline of Mix-of-Show. In single-client concept tuning, the ED-LoRA adopts the layer-wise embedding and multi-word representation. In center node, gradient fusion is adopted to fuse multiple concept LoRAs and then support composing those customized concepts.

### 3.3.1 Single-Client Concept Tuning: ED-LoRA

Vanilla LoRA [3] is not suitable for decentralized multi-concept customization due to the issue of concept conflict. To better understand this limitation, we start by examining the distinct roles of embeddings and LoRA weights in concept tuning.

**Single-Concept Tuning Setting.** We investigate embedding tuning (*i.e.*, Textual Inversion [9] and P+ [4]) and the joint embedding-weight tuning (*i.e.*, LoRA [3]) on single concept customization. We conduct experiments on both in-domain concept (*i.e.*, directly sampled from the pretrained model), and out-domain concepts. The weights of the pretrained model, including the unet $\theta$ and the text encoder $\psi$, are denoted as $\Phi_0 = \{\theta_0, \psi_0\}$. Given a text prompt $P^*$ containing the concept $V$, we visualize the tuned embedding of concept $V$ using the pretrained weights $\Phi_0(P^*)$, and visualize the tuned embedding along with the LoRA weight using $(\Phi_0 + \Delta\Phi)(P^*)$.

**Analysis.** Based on the experiment results in Fig. 3, we draw the following two observations regarding existing embedding tuning and joint embedding-weight tuning approaches.

    **Observation 1**: *The embeddings are capable of capturing concepts within the domain of pretrained models, while the LoRA helps capture out-domain information.*

In Fig. 3(a, b), we observe that embedding tuning approaches such as Textual Inversion and P+ struggle to capture out-domain concepts. This is because they attempt to encode all out-domain details (*e.g.*, anime styles or details not modeled by the pretrained model $\Phi_0$) within the embedding, resulting in semantic collapse. However, for in-domain concepts sampled from the model, embedding tuning accurately encodes the concept identity within the embedding, benefiting from the accurate modeling of concept details by the pretrained model weights $\Phi_0$. Furthermore, when jointly tuning the embedding with LoRA, the embedding no longer produces oversaturated outputs. This is because the out-domain information is captured by the pretrained model with LoRA weight shift (*i.e.*, $\Phi_0 + \Delta\Phi$).

    **Observation 2**: *Existing LoRA weights encode most of the concept identity and project semantically similar embeddings to visually distinct concepts, leading to conflicts during concept fusion.*

In the joint embedding-LoRA tuning results shown in Fig. 3(c), we observe that directly visualizing the embedding with the pretrained model $\Phi_0(P^*)$ yields semantically similar results. However, when the LoRA weights are loaded $(\Phi_0 + \Delta\Phi)(P^*)$, the target concept can be accurately captured. This suggests that the majority of the concept identity is encoded within the LoRA weights rather than the embedding itself. However, when attempting to support multiple semantically similar concepts within a single model, it becomes problematic to determine which concept to sample based on similar embeddings, resulting in concept conflicts. As shown in Fig. 3(e), when fused into one model, the identity of each individual concept is lost.

**Our Solution: ED-LoRA.** Based on the aforementioned observations, our ED-LoRA is designed to preserve more in-domain essence within the embedding while capturing the remaining details using LoRA weights. To achieve this, we enhance the expressiveness of the embedding through decomposed embedding. As illustrated in Fig. 4, we adopt a layer-wise embedding similar to [4] and create a multi-world representation for the concept token ($V = V_{rand}^+ V_{class}^+$). Here, $V_{rand}^+$ is randomly initialized to capture the variance of different concepts, while $V_{class}^+$ is initialized based on

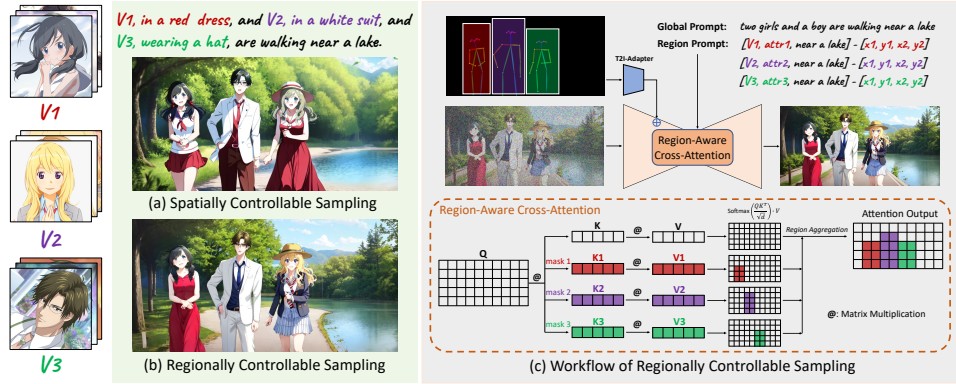

Figure 5: Regionally controllable sampling for multi-concept generation.

its semantic class to maintain semantic meaning. Both tokens are learnable during concept tuning. As shown in Fig. 3(d), the learned embedding of ED-LoRA effectively preserves the essence of the given concept within the domain of the pretrained model, while LoRA helps capture the other details.

### 3.3.2 Center-Node Concept Fusion: Gradient Fusion

At the center node, we have access to all the concept LoRAs and can use these models to update the pretrained model, enabling multi-concept customization. However, the existing weight fusion strategy described in Eq. 2 is insufficient to achieve this goal, as we will discuss further below.

**Multi-Concept Fusion Setting.** In this experiment, we apply the weight fusion described in Eq. 2 to weighted average $n$ concept LoRAs or ED-LoRAs $\{\Delta\Phi_i, i = 1 \cdots n\}$ into the pretrained model $\Phi_0$, resulting in a new model $\Phi$. We then use the new model $\Phi(P_i^*)$ to sample each concept and compare its identity with the corresponding single-concept sample $(\Phi_0 + \Delta\Phi)(P_i^*)$.

**Analysis.** Based on the results in Fig. 3, we make the following observation about fusion strategy.

> **Observation 3**: *Weight fusion leads to identity loss of individual concepts in concept fusion.*

As shown in Fig. 3 (multi-concept), we can observe that weight fusion in the case of LoRA leads to significant loss of concept identity due to conflicts between concepts. Even when combined with our ED-LoRA, weight fusion still compromises the identity of each individual concept. In theory, if a concept achieves its complete identity through LoRA weight shift $\Delta\Phi(P^*)$, fusing it with $n$-1 other concept LoRAs requires reducing its weight to $\frac{1}{n}\Delta\Phi(P^*)$ and introducing other concept LoRA weights, which ultimately diminishes the concept's identity.

**Our Solution: Gradient Fusion.** Based on the previous analysis, our objective is to preserve the identity of each concept in the fused model by aligning their single-concept inference behavior. Unlike in federated learning [17, 19], where models are typically single-direction classification models that cannot access gradients without data, text-to-image diffusion models have the inherent capability to decode concepts from text prompts. Leveraging this characteristic, we first decode the individual concepts using their respective LoRA weights, as depicted in Fig. 4(b). We then extract the input and output features associated with each LoRA layer. These input/output features from different concepts are concatenated, and fused gradients are used to update each layer $W$ using the following objective: $W = \arg\min_W \sum_{i=1}^n ||(W_0 + \Delta W_i)X_i - WX_i||_F^2$, where $X_i$ represents the input activation of the $i$-th concept, and $|\cdot|_F$ denotes the Frobenius norm. By adopting this approach, we can fuse different concept LoRAs without accessing the data and without considering their differences during training. The results of our gradient fusion are shown in Fig. 3(f), demonstrating improved preservation of each concept's identity and consistent stylization across different concepts.

### 3.4 Regionally Controllable Sampling

Direct multi-concept sampling often encounters challenges of missing objects and attribute binding [6, 31, 32, 33, 34]. While spatially controllable sampling methods (*e.g.*, ControlNet [7] and T2I-Adapter [8]) can address the issue of missing objects in multi-concept generation, they cannot accurately bind concepts to specific keyposes or sketches. Merely indicating the desired concept

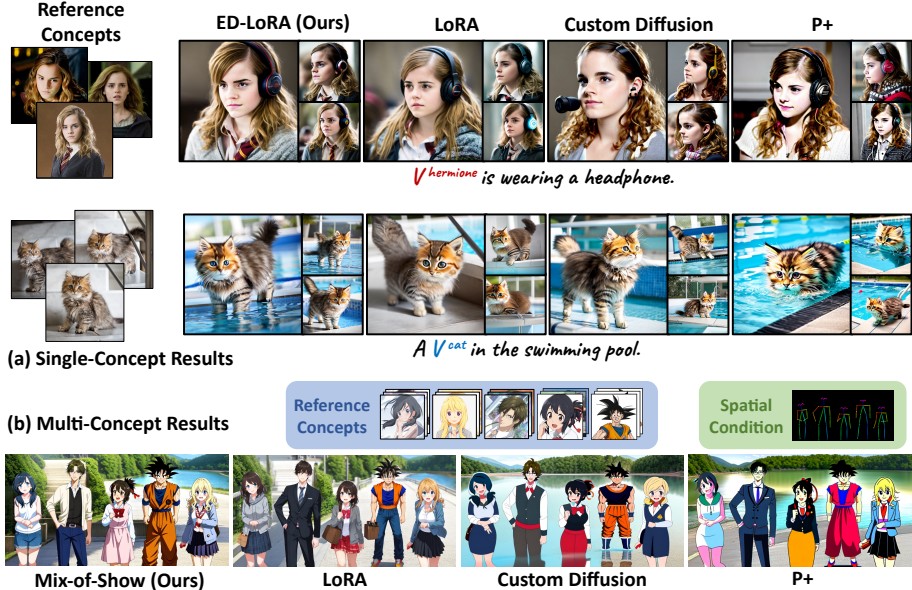

**Figure 6:** Qualitative comparison on single- and multi-concept customization.

| | Methods | Real-Objects Single→Fused | Real-Characters Single→Fused | Real-Scenes Single→Fused | Mean Change |
|---|---|---|---|---|---|
| **Text-alignment** | Upper Bound | 0.811 | 0.767 | 0.834 | 0.804 |
| | P+ [4] | 0.771→0.771 (-) | 0.686→0.686 (-) | 0.759→0.759 (-) | 0.739→0.739 (-) |
| | Custom Diffusion [11] | 0.745→0.747 (+0.002) | 0.674→0.650 (-0.024) | 0.748→0.738 (-0.010) | 0.722→0.712 (-0.010) |
| | LoRA [3] | 0.720→0.795 (+0.075) | 0.654→0.700 (+0.046) | 0.717→0.760 (+0.043) | 0.697→0.752 (+0.055) |
| | Mix-of-Show (Ours) | 0.724→0.745 (+0.021) | 0.632→0.662 (+0.030) | 0.716→0.736 (+0.020) | 0.691→0.714 (+0.024) |
| **Image-alignment** | Lower Bound | 0.721 | 0.471 | 0.595 | 0.596 |
| | P+ [4] | 0.790→0.790 (-) | 0.670→0.670 (-) | 0.796→0.796 (-) | 0.752→0.752 (-) |
| | Custom Diffusion [11] | 0.842→0.808 (-0.034) | 0.714→0.694 (-0.020) | 0.804→0.750 (-0.054) | 0.787→0.751 (-0.036) |
| | LoRA [3] | 0.864→0.778 (-0.086) | 0.761→0.555 (-0.206) | 0.824→0.769 (-0.055) | 0.816→0.701 (-0.115) |
| | Mix-of-Show (Ours) | 0.868→0.846 (-0.022) | 0.802→0.770 (-0.032) | 0.858→0.838 (-0.020) | **0.843→0.818** (-0.025) |

**Table 1:** *Text-alignment* and *image-alignment* vary between the single-client tuned model and the center-node fused model. The upper bound of text-alignment and the lower bound of image-alignment are computed by replacing the concept's token (*e.g.*, $V^{dogA}$) with its class token (*e.g.*, dog) and sampling using the pretrained model.

and attribute through a text prompt can lead to attribute binding problems, as in Fig. 5(a), where the identities of three people are mixed, and the "red dress" is incorrectly assigned to other concepts.

To address these challenges, we propose a method called regionally controllable sampling. This approach utilizes both a global prompt and multiple regional prompts to describe an image based on spatial conditions. The global prompt provides the overall context, while the regional prompts specify subjects within specific regions, including their attributes and contextual information from the global prompt (*e.g.*, "near a lake"). To achieve this, we introduce region-aware cross-attention. Given a global prompt $P_g^*$ and $n$ regional prompts $P_{r_i}^*$, we first incorporate the global prompt via cross-attention with the latent $z$ by $h = \text{softmax}\left(\frac{Q(z)K(P_g^*)}{\sqrt{d}}\right) \cdot V(P_g^*)$. Then, we extract the regional latent feature by $z_i = z \odot M_i$, where $M_i$ represents the binary mask associated with the region specified by $P_{r_i}^*$. We obtain regional features using $h_i = \text{softmax}\left(\frac{Q(z_i)K(P_{r_i}^*)}{\sqrt{d}}\right) \cdot V(P_{r_i}^*)$. Finally, we replace the features in the global output with the regional features: $h[M_i] = h_i$. As shown in Fig. 5(b), regionally controllable sampling allows for precise assignment of subjects and attributes, while maintaining a harmonious global context.

## 4 Experiments

### 4.1 Datasets and Implementation Details

To conduct evaluation for Mix-of-Show, we collect a dataset containing characters, objects, and scenes. For ED-LoRA tuning, we incorporate LoRA layer into the linear layer in all attention

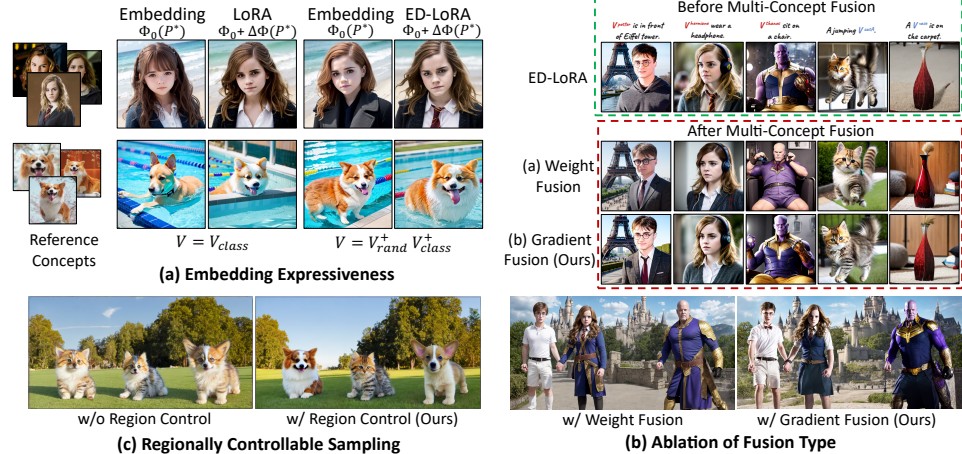

Figure 7: Qualitative ablation study of Mix-of-Show. $P^*$ means the text prompt. $\Phi_0$ and $\Delta\Phi$ denotes the pretrained model and LoRA weight, respectively.

| | Methods | Real-Objects Single→Fused | Real-Characters Single→Fused | Real-Scenes Single→Fused | Mean Change |
|---|---|---|---|---|---|
| | Lower Bound | 0.721 | 0.471 | 0.595 | 0.596 |
| Image-alignment | LoRA + Weight Fusion | 0.864→0.778 (**-0.086**) | 0.761→0.555(**-0.206**) | 0.824→0.769 (**-0.055**) | 0.816→0.701 (**-0.115**) |
| | ED-LoRA + Weight Fusion | 0.868→0.798 (**-0.070**) | 0.802→0.634 (**-0.168**) | 0.858→0.816 (**-0.042**) | 0.843→0.749 (**-0.094**) |
| | ED-LoRA + Gradient Fusions | 0.868→0.846 (**-0.022**) | 0.802→0.770 (**-0.032**) | 0.858→0.838 (**-0.020**) | **0.843→0.818** (**-0.025**) |

(a) Subject identity preservation (*i.e., image-alignment*) measured by CLIP score between LoRA+weight fusion, ED-LoRA+weight fusion, and ED-LoRA+gradient fusion. Our ED-LoRA+gradient fusion achieves the least loss in image-alignment after multi-concept fusion, preserving the best subject identity.

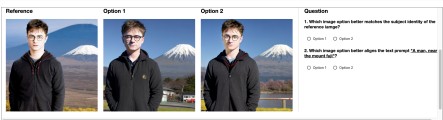

| Human Evaluation | Image Alignment | Text Alignment |
|---|---|---|
| ED-LoRA + Weight Fusion | 33.5% | 47.5% |
| ED-LoRA + Gradient Fusion | **66.5%** | **52.5%** |

(b) Human preference study interface on Amazon Mechanical Turk.

(c) Human preference study between weight fusion and gradient fusion for fusing ED-LoRAs.

Table 2: Quantitative ablation study of our main components. (a) Ablation study between the LoRA+weight fusion, ED-LoRA+weight fusion and our ED-LoRA+gradient fusion. (b, c) Human preference study to comparing weight fusion and gradient fusion for fusing our ED-LoRAs.

modules of the text encoder and Unet, with a rank of $r = 4$ in all experiments. We use the Adam [48] optimizer with a learning rate of 1e-3, 1e-5 and 1e-4 for tuning text embedding, text encoder and Unet, respectively. For gradient fusion, we use the LBFGS optimizer [49] with 500 and 50 steps to optimize the text encoder and Unet, respectively. More details are provided *in the supplementary*.

## 4.2 Qualitative Comparison

**Single-Concept Results.** We compare our ED-LoRA with LoRA [3], Custom Diffusion [11] and P+ [4] for single-concept customization. The results are shown in Fig. 6 (a). Our ED-LoRA achieves comparable performance to previous methods on customizing objects, while maintaining better identity for character customization. More comparisons are provided *in the supplementary*.

**Multi-Concept Results.** We compare Mix-of-Show with LoRA [3], Custom Diffusion [11], and P+ [4] for decentralized multi-concept customization. For LoRA,we utilize weight fusion to combine the different concepts. In the case of P+, we directly incorporate the tuned concept embedding into the pretrained model. And for Custom Diffusion, we follow their approach of constrained optimization to merge the key and value projections in cross-attention. To ensure fair evaluation, we employ the same regionally controllable sampling for multi-concept generation across all models and the results are summarized in Fig. 6 (b).

P+ [4] and Custom Diffusion [11] only tunes text-related module (*i.e.*, text embedding, or the key and value projection of cross-attention). In contrast, LoRA and Mix-of-Show add LoRA layers to the

entire model. The limited scope of tuned modules in P+ and Custom Diffusion leads to an excessive encoding of out-domain low-level details within the embedding. This leads to unnatural and less desirable outcomes when compared to LoRA and Mix-of-Show. In comparison to LoRA, which loses concept identity after weight fusion, Mix-of-Show effectively preserves the identity of each individual concept.

### 4.3 Quantitative Comparison

Following Custom Diffusion [11], we utilize the CLIP [47] text/image encoder to assess text alignment and image alignment. We evaluate on different category of concepts on both single-concept tuned model and multi-concept fused model. We include detailed evaluation setting *in the supplementary*.

Based on the results presented in Table. 1, both Mix-of-Show and LoRA exhibit superior image alignment compared to other methods, all the while maintaining comparable text alignment in the ***single-client tuned model***. This achievement stems from their fine-tuning the spatial-related layer in Unet (*e.g.*, linear projection layer in self-attention), a critical aspect for accurately capturing the complex concepts' identity, such as characters.

However, the main difference between LoRA and Mix-of-Show emerges in the context of multi-concept fusion. In the ***center-node fused model***, LoRA experiences a significant decline in image alignment for each concept, progressively deteriorating towards the lower bound. In contrast, our Mix-of-Show method undergoes far less degradation in image alignment after multi-concept fusion.

### 4.4 Ablation Study

**Embedding Expressiveness.** In Fig. 7(a), it is evident that our decomposed embeddings better preserve the identity of the specified concept compared to the standard text embeddings used in LoRA. This results in a more robust encoding of concept identity. As shown in the quantitative results in Table. 2(a), when LoRA is replaced with ED-LoRA, the identity loss from weight fusion (measured by mean change of image-alignment) is reduced from 0.115 to 0.094. This result verifies that expressive embeddings help reduce identity loss during multi-concept fusion.

**Fusion Type.** Built with the same ED-LoRAs, we conduct experiments to compare weight fusion and gradient fusion. As shown in Fig. 7(b), gradient fusion effectively preserves concept identity after concept fusion, resulting in superior results for multi-concept sampling. According to the quantitative results in Table. 2(a), gradient fusion significantly reduces the identity loss of weight fusion, decreasing it from 0.094 to 0.025. We also conduct a human evaluation and confirm a clear preference for gradient fusion, as indicated in Table. 2(c).

**Regionally Controllable Sampling.** As shown in Fig. 7(c), direct sampling lead to attribute binding issues, where the concept identities are mixed. However, our regionally controllable sampling overcomes this problem and achieves correct attribute binding in multi-concept generation.

## 5 Conclusion

In this work, we explore decentralized multi-concept customization and highlight the limitations of existing methods like LoRA tuning and weight fusion, which suffer from concept conflicts and identity loss in this scenario. To overcome these challenges, we propose Mix-of-Show, a framework that combines ED-LoRA for single-client concept tuning and gradient fusion for center-node concept fusion. ED-LoRA preserves individual concept essence in the embedding, avoiding conflicts, while gradient fusion minimizes identity loss during concept fusion. We also introduce regionally controllable sampling to handle attribute binding in multi-concept generation. Experiments demonstrate Mix-of-Show can successfully generate complex compositions of multiple customized concepts, including characters, objects and scenes.

## Acknowledgements

This project is supported by the National Research Foundation, Singapore under its NRFF Award NRF-NRFF13-2021-0008, and the Ministry of Education, Singapore, under the Academic Research Fund Tier 1 (FY2022).

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
