# Supplementary Material for '*Mix-of-Show*'

**Yuchao Gu[1], Xintao Wang[3], Jay Zhangjie Wu[1], Yujun Shi[2], Yunpeng Chen[2],
Zihan Fan[2], Wuyou Xiao[2], Rui Zhao[1], Shuning Chang[1], Weijia Wu[1],
Yixiao Ge[3], Ying Shan[3], Mike Zheng Shou[1]**[*]

[1]Show Lab, [2]National University of Singapore    [3]ARC Lab, Tencent PCG

https://showlab.github.io/Mix-of-Show

## Contents

---

[*]Corresponding Author.

37th Conference on Neural Information Processing Systems (NeurIPS 2023).

# 1 Dataset and Implementation Details

## 1.1 Dataset Details

Previous works, such as Dreambooth [1] and Custom Diffusion [2], have primarily focused on object customization. In contrast, our research encompasses a more comprehensive investigation that involves characters, objects, and scenes. To facilitate our experiments, we curate a dataset comprising 19 different concepts, including 6 real-world characters, 5 anime characters, 6 real-world objects, and 2 real-world scenes. The object part is borrowed from Dreambooth [1] and Custom Diffusion [2].

## 1.2 Implementation Details

**Pretrained Models.** Due to the well-known quality issues associated with Stable-Diffusion v1-5 on human faces, we adopt the Chilloutmix[2] as the pretrained model for real-world concepts. Additionally, we employ the Anything-v4[3] as the pretrained model for anime concepts. To ensure fair comparisons with other methods, we run all comparison methods using the same pretrained model.

**Single-Client Concept Tuning.** In the implementation of ED-LoRA tuning, we incorporate LoRA layers into the linear layers of all attention modules within the text encoder and Unet. Throughout our experiments, we maintain a consistent rank of $r = 4$ for the LoRA layers for simplicity. To optimize the different components, we utilize the Adam optimizer [3] with specific learning rates: 1e-3 for text embedding, 1e-5 for the text encoder, and 1e-4 for the Unet. We use a 0.01 noise offset for all experiments, which we have found to be crucial for encoding stable identity.

**Center-Node Concept Fusion.** In the center-node concept fusion, we apply layer-wise optimization to the layer connected with the LoRA layer. Each layer for optimization is initialized with pretrained weights, and we use the LBFGS optimizer [4]. In detail, we optimize the text encoder layer through 500 steps, while the Unet layer requires 50 steps for optimization.

**Sample Details.** In all the experiments and evaluations conducted in this paper, we utilize the DPM-Solver [5] with 20 sampling steps. To filter out undesired variations in diffusion models, we employ the same negative prompt for both our and the comparison methods during sampling: "longbody, lowres, bad anatomy, bad hands, missing fingers, extra digit, fewer digits, cropped, worst quality, low quality."

**Running Times.** In single-client concept tuning, the process of tuning each concept takes approximately 10-20 minutes on two Nvidia-A100 GPUs, taking into account variations in data volume. As for the center-node concept fusion, it takes 30 minutes on a single Nvidia-A100 GPU to merge 14 concepts within the pretrained model.

# 2 Quantitative and Qualitative Evaluation

## 2.1 Evaluation Setting

Our evaluation focuses on investigating the each concept in the **single-concept tuned model** and **the center-node fused model**. To assess the performance, we employ the evaluation metric, which includes image-alignment and text-alignment, as outlined in Custom Diffusion [2]. Specifically, for text-alignment, we evaluates the text-image similarity of the sampled image with the corresponding sample prompt in the CLIP feature space [6] by CLIP-Score toolkit[4]. For image-alignment, we evaluate the pairwise image similarity between the sampled image and the target concept data in the CLIP Image feature space [6].

For each concept, we utilize 20 evaluation prompts, which can be roughly categorized into four types: Recontextualization, Restylization, Interaction, and Property Modification. In Recontextualization, we assess the concept's performance by changing its context to different settings, such as the Eiffel Tower or Mount Fuji. In Restylization, we explore the concept's ability to adapt to various artistic styles. In Interaction, we investigate the concept's capability to interact with other objects, such as

---

[2]https://civitai.com/models/6424/chilloutmix
[3]https://huggingface.co/andite/anything-v4.0/tree/main
[4]https://github.com/jmhessel/clipscore

Figure 1 table:

| | prompts for characters | prompts for pets | prompts for table |
|---|---|---|---|
| **Recontextualization** | A photo of <TOK> on the beach, small waves, detailed symmetric face, beautiful composition
A <TOK>, in front of Eiffel tower
A <TOK>, near the mount fuji
A <TOK>, in the forest
A <TOK>, walking on the street | A <TOK>, in the swimming pool
A <TOK>, in front of Eiffel tower
A <TOK>, near the mount fuji
A <TOK>, in the forest
A <TOK>, walking on the street | A <TOK>, in the swimming pool
A <TOK>, in front of Eiffel tower
A <TOK>, near the mount fuji
A <TOK>, in the forest
A <TOK>, walking on the street |
| **Restylization** | A <TOK>, cyberpunk 2077, 4K, 3d render in unreal engine
A watercolor painting of a <TOK>
A painting of a <TOK> in the style of Vincent Van Gogh
A painting of a <TOK> in the style of Claude Monet
A <TOK> in the style of Pixel Art | A <TOK>, cyberpunk 2077, 4K, 3d render in unreal engine
A watercolor painting of a <TOK>
A painting of a <TOK> in the style of Vincent Van Gogh
A painting of a <TOK> in the style of Claude Monet
A <TOK> in the style of Pixel Art | A <TOK>, cyberpunk 2077, 4K, 3d render in unreal engine
A watercolor painting of a <TOK>
A painting of a <TOK> in the style of Vincent Van Gogh
A painting of a <TOK> in the style of Claude Monet
A <TOK> in the style of Pixel Art |
| **Interaction** | A <TOK> sit on the chair
A <TOK> ride a horse
A <TOK>, wearing a headphone
A <TOK>, wearing a sunglass
A <TOK>, wearing a Santa hat | A <TOK> sit on the chair
A <TOK> on the boat
A <TOK>, wearing a headphone
A <TOK>, wearing a sunglass
A <TOK> playing with a ball | A <TOK> sit on the chair
A <TOK> on the boat
A <TOK>, wearing a headphone
A <TOK>, wearing a sunglass
A <TOK> playing with a ball |
| **Property Change** | A smiling <TOK>
An angry <TOK>
A running <TOK>
A jumping <TOK>
A <TOK> is lying down | A sad <TOK>
An angry <TOK>
A running <TOK>
A jumping <TOK>
A <TOK> is lying down | A sad <TOK>
An angry <TOK>
A running <TOK>
A jumping <TOK>
A <TOK> is lying down |

| | prompts for chair | prompts for vase | prompts for scene |
|---|---|---|---|
| **Recontextualization** | A <TOK>, in the snow
A <TOK>, at night
A <TOK>, in autumn
A <TOK>, in a sunny day
A <TOK>, in thunder and lightning | A <TOK>, in the snow
A <TOK>, at night
A <TOK>, in autumn
A <TOK>, in a sunny day
A <TOK>, in thunder and lightning | A <TOK>, in the snow
A <TOK>, at night
A <TOK>, in autumn
A <TOK>, in a sunny day
A <TOK>, in thunder and lightning |
| **Restylization** | A <TOK>, cyberpunk 2077, 4K, 3d render in unreal engine
A watercolor painting of a <TOK>
A painting of a <TOK> in the style of Vincent Van Gogh
A painting of a <TOK> in the style of Claude Monet
A <TOK> in the style of Pixel Art | A <TOK>, cyberpunk 2077, 4K, 3d render in unreal engine
A watercolor painting of a <TOK>
A painting of a <TOK> in the style of Vincent Van Gogh
A painting of a <TOK> in the style of Claude Monet
A <TOK> in the style of Pixel Art | A <TOK>, cyberpunk 2077, 4K, 3d render in unreal engine
A watercolor painting of a <TOK>
A painting of a <TOK> in the style of Vincent Van Gogh
A painting of a <TOK> in the style of Claude Monet
A <TOK> in the style of Pixel Art |
| **Interaction** | A girl near the <TOK>
A boy near the <TOK>
A dog near the <TOK>
A cat near the <TOK>
Many people near the <TOK> | A girl near the <TOK>
A boy near the <TOK>
A dog near the <TOK>
A cat near the <TOK>
Many people near the <TOK> | A girl near the <TOK>
A boy near the <TOK>
A dog near the <TOK>
A cat near the <TOK>
Many people near the <TOK> |
| **Property Change** | A <TOK> in rainbow colors
A <TOK> made of metal
A close view of <TOK>
A top view of <TOK>
A bottom view of <TOK> | A <TOK> in rainbow colors
A <TOK> made of metal
A close view of <TOK>
A top view of <TOK>
A bottom view of <TOK> | A <TOK> in rainbow colors
A <TOK> made of metal
A close view of <TOK>
A top view of <TOK>
A bottom view of <TOK> |

Figure 1: Summarization of our evaluation prompts for each concept.

associations or actions like sitting on a chair. In Property Modification, we modify the internal state of the concept, including expressions or states like running or jumping. Each type consists of 5 prompts, some of which are borrowed from previous work, resulting in a total of 20 prompts per concept. We sample 50 images for each prompt, ensuring reproducibility by fixing the random seed within the range of [1, 50]. This yields a total of 1000 images for each concept. The evaluation prompts for each concept are presented in Fig. 1.

## 2.2 Quantitative Results

According to the evaluation setting described in Sec. 2.1, we have compiled the complete evaluation results for each concept, which are summarized in Table. 1. The summarized results of different categories can be found in Table. 1 of the main paper.

## 2.3 Qualitative Results

The qualitative comparison of Mix-of-Show and other methods on the single-client tuned model and center-node fused model is presented in Fig. 2. From the results, it is evident that the LoRA experiences the most significant loss of concept identity after concept fusion. Additionally, due to the limited tunability of positions in the P+ and Custom Diffusion, they exhibit oversaturated results or semantic collapse in some examples. Conversely, Mix-of-Show consistently achieves the best concept identity and quality across various examples, while also minimizing the loss of identity after the center-node concept fusion.

# 3 Limitation and Future Work

## 3.1 Limitation

The first limitation is related to regionally controllable sampling, as depicted in Fig. 3(a), where attributes from one region may influence another due to the encoding of some attributes in the global

| Single-Concept Model | Methods | Cat (5) | DogA (5) | Chair (5) | Table (4) | DogB (5) | Vase (6) | Mean |
|---|---|---|---|---|---|---|---|---|
| Text-alignment | Upper Bound | 0.832 | 0.821 | 0.795 | 0.792 | 0.821 | 0.807 | 0.811 |
| | P+ [7] | 0.828 | 0.801 | 0.726 | 0.696 | 0.799 | 0.776 | 0.771 |
| | Custom Diffusion [2] | 0.784 | 0.744 | 0.698 | 0.689 | 0.786 | 0.768 | 0.745 |
| | LoRA [8] | 0.761 | 0.673 | 0.655 | 0.670 | 0.779 | 0.779 | 0.720 |
| | Mix-of-Show (Ours) | 0.771 | 0.703 | 0.666 | 0.671 | 0.772 | 0.759 | 0.724 |
| Image-alignment | Lower Bound | 0.753 | 0.755 | 0.674 | 0.682 | 0.769 | 0.692 | 0.721 |
| | P+ [7] | 0.783 | 0.753 | 0.809 | 0.810 | 0.825 | 0.761 | 0.790 |
| | Custom Diffusion [2] | 0.869 | 0.830 | 0.825 | 0.888 | 0.848 | 0.794 | 0.842 |
| | LoRA [8] | 0.859 | 0.871 | 0.872 | 0.917 | 0.887 | 0.778 | 0.864 |
| | Mix-of-Show (Ours) | 0.874 | 0.864 | 0.890 | 0.879 | 0.889 | 0.811 | **0.868** |

| Fused Model | Methods | Cat (5) | DogA (5) | Chair (5) | Table (4) | DogB (5) | Vase (6) | Mean |
|---|---|---|---|---|---|---|---|---|
| Text-alignment | Upper Bound | 0.832 | 0.821 | 0.795 | 0.792 | 0.821 | 0.807 | 0.811 |
| | P+ [7] | 0.828 | 0.801 | 0.726 | 0.696 | 0.799 | 0.776 | 0.771 |
| | Custom Diffusion [2] | 0.756 | 0.735 | 0.726 | 0.724 | 0.782 | 0.758 | 0.747 |
| | LoRA [8] | 0.827 | 0.803 | 0.757 | 0.766 | 0.814 | 0.801 | 0.795 |
| | Mix-of-Show (Ours) | 0.801 | 0.738 | 0.673 | 0.709 | 0.786 | 0.761 | 0.745 |
| Image-alignment | Lower Bound | 0.753 | 0.755 | 0.674 | 0.682 | 0.769 | 0.692 | 0.721 |
| | P+ [7] | 0.783 | 0.753 | 0.809 | 0.810 | 0.825 | 0.761 | 0.790 |
| | Custom Diffusion [2] | 0.871 | 0.813 | 0.774 | 0.794 | 0.823 | 0.773 | 0.808 |
| | LoRA [8] | 0.805 | 0.800 | 0.761 | 0.778 | 0.808 | 0.715 | 0.778 |
| | Mix-of-Show (Ours) | 0.852 | 0.863 | 0.864 | 0.816 | 0.880 | 0.800 | **0.846** |

(a) Quantitative results from single-concept model and center-node fused model on **real-world objects**.

| Single-Concept Model | Methods | Potter (14) | Hermione (15) | Thanos (15) | Hinton (14) | Lecun (17) | Bengio (15) | Mean |
|---|---|---|---|---|---|---|---|---|
| Text-alignment | Upper Bound | 0.765 | 0.776 | 0.765 | 0.765 | 0.765 | 0.765 | 0.767 |
| | P+ [7] | 0.640 | 0.744 | 0.622 | 0.708 | 0.693 | 0.711 | 0.686 |
| | Custom Diffusion [2] | 0.654 | 0.720 | 0.594 | 0.717 | 0.683 | 0.674 | 0.674 |
| | LoRA [8] | 0.580 | 0.696 | 0.568 | 0.716 | 0.681 | 0.684 | 0.654 |
| | Mix-of-Show (Ours) | 0.575 | 0.650 | 0.562 | 0.665 | 0.680 | 0.662 | 0.632 |
| Image-alignment | Lower Bound | 0.485 | 0.458 | 0.510 | 0.422 | 0.510 | 0.441 | 0.471 |
| | P+ [7] | 0.778 | 0.608 | 0.809 | 0.582 | 0.614 | 0.629 | 0.670 |
| | Custom Diffusion [2] | 0.737 | 0.663 | 0.852 | 0.627 | 0.694 | 0.710 | 0.714 |
| | LoRA [8] | 0.866 | 0.679 | 0.917 | 0.683 | 0.716 | 0.705 | 0.761 |
| | Mix-of-Show (Ours) | 0.869 | 0.785 | 0.921 | 0.731 | 0.723 | 0.782 | **0.802** |

| Fused Model | Methods | Potter (14) | Hermione (15) | Thanos (15) | Hinton (14) | Lecun (17) | Bengio (15) | Mean |
|---|---|---|---|---|---|---|---|---|
| Text-alignment | Upper Bound | 0.765 | 0.776 | 0.765 | 0.765 | 0.765 | 0.765 | 0.767 |
| | P+ [7] | 0.640 | 0.744 | 0.622 | 0.708 | 0.693 | 0.711 | 0.686 |
| | Custom Diffusion [2] | 0.604 | 0.678 | 0.624 | 0.699 | 0.651 | 0.641 | 0.650 |
| | LoRA [8] | 0.693 | 0.717 | 0.656 | 0.694 | 0.714 | 0.725 | 0.700 |
| | Mix-of-Show (Ours) | 0.632 | 0.677 | 0.611 | 0.673 | 0.678 | 0.701 | 0.662 |
| Image-alignment | Lower Bound | 0.485 | 0.458 | 0.510 | 0.422 | 0.510 | 0.441 | 0.471 |
| | P+ [7] | 0.778 | 0.608 | 0.809 | 0.582 | 0.614 | 0.629 | 0.670 |
| | Custom Diffusion [2] | 0.765 | 0.680 | 0.749 | 0.603 | 0.693 | 0.671 | 0.694 |
| | LoRA [8] | 0.558 | 0.600 | 0.792 | 0.412 | 0.523 | 0.447 | 0.555 |
| | Mix-of-Show (Ours) | 0.827 | 0.756 | 0.867 | 0.710 | 0.729 | 0.733 | **0.770** |

(b) Quantitative results from single-concept model and center-node fused model on **real-world characters**.

| Single-Concept Model | Methods | Rock (20) | Pyramid (20) | Mean |
|---|---|---|---|---|
| Text-alignment | Upper Bound | 0.869 | 0.798 | 0.834 |
| | P+ [7] | 0.801 | 0.716 | 0.759 |
| | Custom Diffusion [2] | 0.809 | 0.686 | 0.748 |
| | LoRA [8] | 0.737 | 0.697 | 0.717 |
| | Mix-of-Show (Ours) | 0.754 | 0.677 | 0.716 |
| Image-alignment | Lower Bound | 0.672 | 0.517 | 0.595 |
| | P+ [7] | 0.821 | 0.770 | 0.796 |
| | Custom Diffusion [2] | 0.808 | 0.800 | 0.804 |
| | LoRA [8] | 0.863 | 0.784 | 0.824 |
| | Mix-of-Show (Ours) | 0.859 | 0.857 | **0.858** |

| Fused Model | Methods | Rock (20) | Pyramid (20) | Mean |
|---|---|---|---|---|
| Text-alignment | Upper Bound | 0.869 | 0.798 | 0.834 |
| | P+ [7] | 0.801 | 0.716 | 0.759 |
| | Custom Diffusion [2] | 0.793 | 0.682 | 0.738 |
| | LoRA [8] | 0.786 | 0.734 | 0.760 |
| | Mix-of-Show (Ours) | 0.770 | 0.702 | 0.736 |
| Image-alignment | Lower Bound | 0.672 | 0.517 | 0.595 |
| | P+ [7] | 0.821 | 0.770 | 0.796 |
| | Custom Diffusion [2] | 0.742 | 0.757 | 0.750 |
| | LoRA [8] | 0.810 | 0.728 | 0.769 |
| | Mix-of-Show (Ours) | 0.832 | 0.844 | **0.838** |

(c) Quantitative results from single-concept model and center-node fused model on **real-world scenes**.

Table 1: *Text-alignment* and *image-alignment* of the single-client tuned model and the center-node fused model. The upper bound of text-alignment and the lower bound of image-alignment are computed by replacing the concept's token with its class token and sampling using the pretrained model. For instance, to assess the upper-bound text alignment and lower-bound image alignment for the concept "$V^{dogA}$," we substitute "$V^{dogA}$" in the sample prompts with the class token "dog" and sample it using the pretrained model. ($N$) means each concept has $N$ images for tuning.

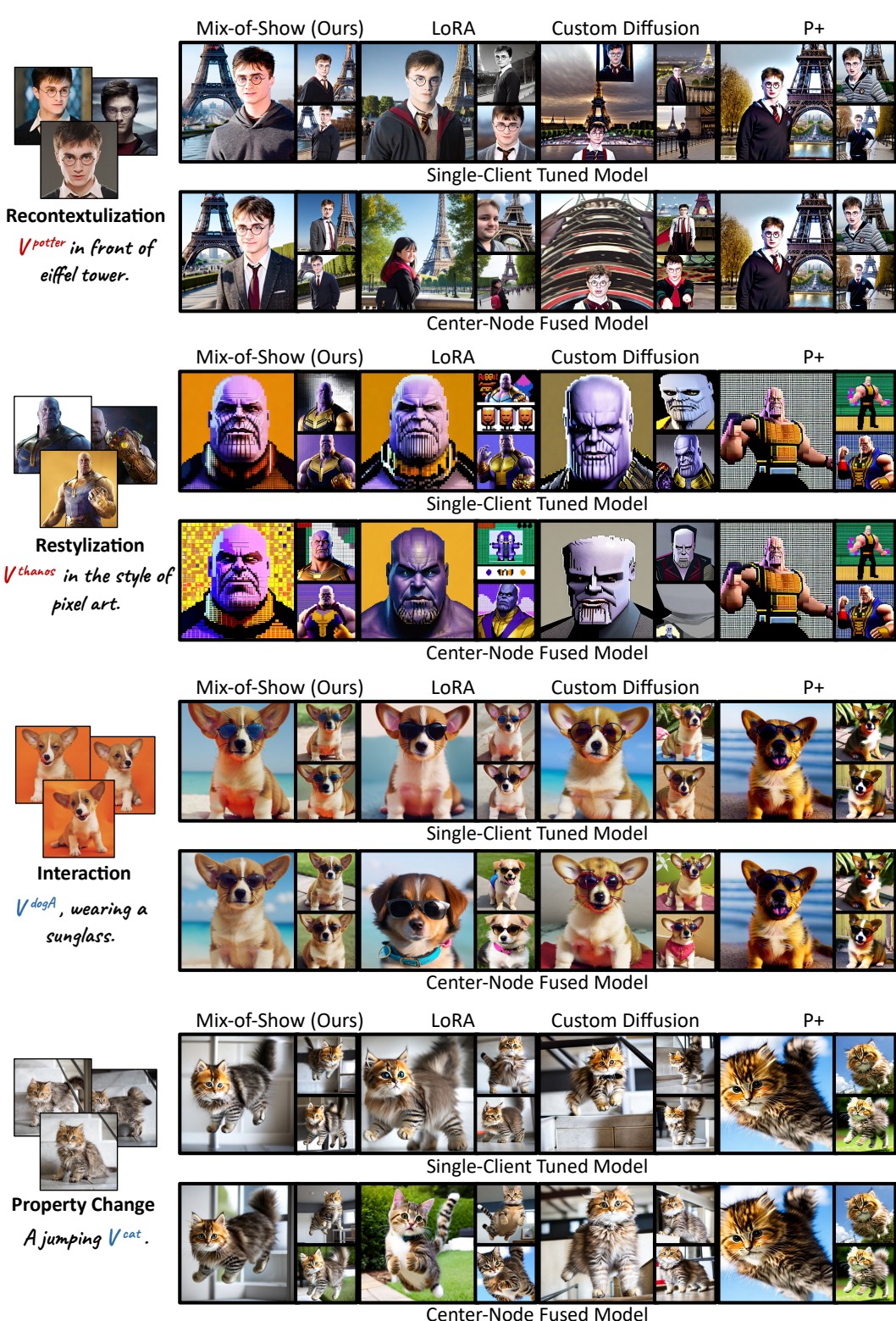

Figure 2: Qualitative comparison of Mix-of-Show *vs.* P+ [7], Custom Diffusion [2], and LoRA [8]. Our Mix-of-Show demonstrates superior concept identity and quality in single-concept tuned model and achieves the least identity loss after concept fusion.

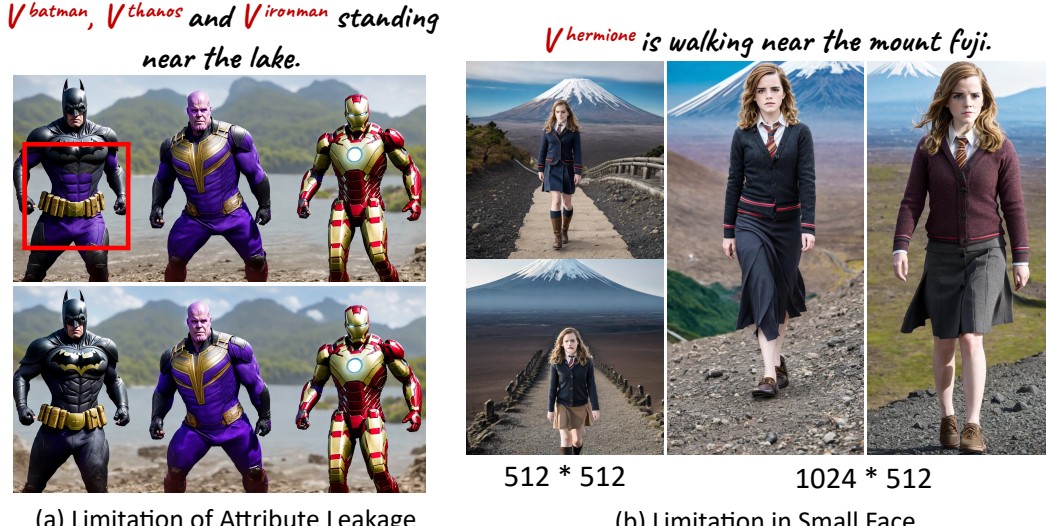

*V*batman, *V*thanos and *V*ironman standing near the lake.

*V*hermione is walking near the mount fuji.

512 * 512       1024 * 512

(a) Limitation of Attribute Leakage       (b) Limitation in Small Face

Figure 3: Limitation of Mix-of-Show. (a) Attribute leakage in regionally controllable sampling. (b) Limitation in small face generation.

embedding. This issue can be partially alleviated by specifying undesired attributes using negative prompts for each region.

The second limitation concerns the center-node concept fusion, which requires a relatively lengthy time to merge concepts. The primary bottleneck in this process arises from the presence of large spatial features in the Unet layer during layer-wise optimization.

The final limitation relates to the generation of small faces. In the case of Stable Diffusion, the information loss in the VAE can affect the generation of high-quality full-body characters, especially in the small face region, resulting in a loss of facial details, as shown in Fig. 3(b). To mitigate this limitation, increasing the sample size can be a potential solution.

## 3.2 Future Work

Our Mix-of-Show framework enables the reusability and scalability of tuned concepts, facilitating the creation of complex multi-concept compositions. In future work, it would be interesting to explore how Mix-of-Show can enhance storybook generation by generating character and object interactions across various plots. Furthermore, as Mix-of-Show supports stable identity encoding, it has the potential to assist in concept customization for video or 3D scenarios.

## 3.3 Potential Negative Society Impact

This project aims to provide the community with an effective tool for decentralized creation of high-quality customized concept models and the ability to reuse and combine different concepts to compose complex images. However, a risk exists wherein malicious entities could exploit this framework to create deceptive interactions with real-world figures, potentially misleading the public. This concern is not unique to our approach but rather a shared consideration in other multi-concept customization methodologies. One potential solution to mitigate such risks involves adopting methods similar to anti-dreambooth [9], which introduce subtle noise perturbations to the published images to mislead the customization process. Additionally, applying unseen watermarking to the generated images could deter misuse and prevent them from being used without proper recognition.