# OpenReview forum: "Mix-of-Show: Decentralized Low-Rank Adaptation for Multi-Concept Customization of Diffusion Models"
_NeurIPS.cc/2023/Conference — NeurIPS 2023 poster_

### Official Review · Reviewer_NQJc · 2023-06-19

**Soundness:** 3 good
**Presentation:** 3 good
**Contribution:** 3 good
**Rating:** 6
**Confidence:** 4

**Summary:**

This paper proposes Mix-of-Show for decentralized multi-concept customization using diffusion models. To tackle the concept conflict and identity loss problem, the authors propose the embedding-decomposed LoRA and the gradient fusion. The regionally controllable sampling is proposed to address the attribute binding and missing object problems. Experiments show that the proposed method outperforms existing methods in multi-concept customization.

**Strengths:**

1. Most part of the paper is well-written and easy to understand.

2. Each component of the proposed method is reasonable and well-motivated.

3. The three observations are quite important and interesting to finetuning pretrained diffusion models.

4. Experiments show that the proposed Mix-of-Show outperforms existing methods in multi-concept customization.

**Weaknesses:**

1. The gradient fusion is also adopted in CustomFusion, it is better to clarify it in the paper.

2. The biggest concern is the necessity of the decentralized design, since in current customization scenario, we can always obtain all the subjects together and we can fine-tune them all, and also the CustomFusion indicates this leads to better results. If the decentralization is not necessary, the contribution of this paper will become smaller.

3. Some minor typos, e.g., the caption for Figure 2(b) seems to be incorrect.

**Questions:**

Please refer to the weakness part. It is better to emphasize the necessity of the decentralized design.

**Limitations:**

The authors discussed the limitations in the appendix.

---

> ### Author Rebuttal · Authors · 2023-08-04
>
> Thanks for acknowledging our work. The concerns are addressed as follows:
>
> **Notation**: W: Weakness
>
> ---
>
> **W.1**: Thanks for raising this concern. Both the Custom Diffusion [1] and Mix-of-Show formulate multi-concept fusion as an optimization problem, and thus both methods are indeed related to the gradient. However, the difference lies in how they handle the optimization problem. Custom Diffusion formulates it as a **constrained optimization** problem, where the gradient needs to be considered from both the new concepts (i.e., optimization target) and old concepts (i.e., constraints). On the other hand, Mix-of-Show formulates it as a **non-constrained optimization** problem, which only requires fusing the gradient from new concepts.
>
> Our non-constrained optimization formulation **eases the difficulties in handling constraints** compared to Custom Diffusion. Custom Diffusion derived a closed-form solution for the constrained optimization using Lagrange multipliers, but it only supports merging key and value projection layers of cross-attention, where the input feature is the text embedding. However, for layers where the input contains spatial features, such as query, key, and value projections of self-attention, it does not work because:
>
> - The spatial feature is significantly larger than text embedding, which makes the problem scale too large to compute the closed-form solution.
> - Unlike text embedding, spatial features are usually redundant, leading to redundant constraints, which hinder the computation/accuracy of the Lagrange multiplier.
>
> However, in mix-of-show, we don't need to handle constraints because, in our ED-LoRA, we have learned **concept embeddings that already encode the essence of the concept**, which will not conflict with existing embeddings. Therefore, in merging multiple concepts, the changed weight only relates to the new embedding, and thus we can remove the constraints and formulate the problem as a non-constrained optimization. Without the restriction of constraints, we can **flexibly merge enlarged concepts without concept conflicts** and **support the merging of different modules**, including cross-attention, self-attention, and text encoders, making it highly compatible with LoRAs.
>
> We will incorporate these discussions to clarify the similarity and difference between Custom Diffusion and Mix-of-Show in the revised version of the paper.
>
> ---
>
> **W.2**: Thanks for raising this concern. The decentralized setting is necessary for achieving **reusability and scalability**, **avoiding feature mixing**, and **facilitating tuning and model selection**.
>
> - **Reusability and Scalability**: Considering a real scenario that users share tuned models instead of sharing their collected source data. For instance, they may share LoRA models based on Stable Diffusion on platforms like Civitai [1] or Huggingface [2]. In such a scenario, if we want to make our **own concept interact with the shared concept**, joint tuning doesn't work since it cannot access to source data of shared concepts. Although users can fine-tune their customized concept data on top of the shared model, it probably leads to gradual degradation of the performance of previously learned concepts. This phenomenon is known as catastrophic forgetting in continuous learning. However, in our decentralized setting, these issues are naturally overcome. We can direct reuse the share concept models without source data.
>
> Even in a setting where all customized concept data is available, decentralized multi-concept customization (i.e., separate tuning and then model fusion) offers a better solution because:
>
> - **Mitigating the feature mixing issue**: As mentioned in SVDiff [3], when Custom Diffusion [4] fine-tunes multiple concepts together, features from the similar category can become mixed. This was also the issue encountered in our preliminary exploration. On the other hand, Mix-of-Show performs separate tuning, which first gets the embeddings describe the essence of given concept. The learned LoRA weight only related to those embeddings, mitigating feature mixing with other concepts in joint tuning.
>
> - **Reduce difficulties in evaluating all concepts together**. Users will need to tweak their dataset repeat number or tuning epochs on each concept to select the best performing checkpoints. Balancing the training epochs of different concepts and achieving a good checkpoint considering the quality of all concepts in joint training becomes problematic when the concept numbers increasing and the data number of each concept varies. This problem is not mentioned in Custom Diffusion [4] or SVDiff [3] since they focus on customize 2 object concepts. But it is problematic in our case, i.e., customizing 14 concepts with varying number of different concepts (about 15 images for character, 5 for objects and 20 for scenes).
>
> In summary, decentralized multi-concept customization mitigating feature mixing, reduces the workload of jointly evaluating all new concept performance in concept tuning, and becomes the most flexible and reusable way to scale up concept customizations for production.
>
> ---
>
> **W.3**: Thanks for pointing out, in the Figure 2(b), the caption should be “a $V^{dogA}$, a $V^{cat}$ and a $V^{dogB}$, on the grass, under the sunset”.
>
> ---
>
> **Reference**:
>
> [1] https://civitai.com/
>
> [2] https://huggingface.co/
>
> [3] SVDiff: Compact Parameter Space for Diffusion Fine-Tuning. ICCV 2023.
>
> [4] Multi-Concept Customization of Text-to-Image Diffusion. CVPR 2023.

---

> > ### Comment · Reviewer_NQJc · 2023-08-19
> >
> > Thanks for the author's rebuttal. Most of my concerns are addressed. There is still a minor concern. Since this paper combines many techniques/tricks, the effectiveness of each part is really hard to validate, e.g., in Figure 7 in the appendix, it's hard to understand why without the region control, the subjects will become mixed. Considering the overall project performance, I keep the borderline accept as my final rating.

---

> > > ### Author Response · Authors · 2023-08-19
> > > **Further Response to Reviewer NQJc**
> > >
> > > Dear Reviewer NQJc:
> > >
> > > Thanks for your reply.
> > >
> > > In Figure 7, without our regionally-controllable sampling, the vanilla stable diffusion sampling encounters the attribute binding problem, as discussed in "Training-Free Structure Diffusion Guidance" [1]. This problem is mainly caused by CLIP text encoding, which mixes different adjectives with nouns because of contextualize embedding. However, with our regionally controllable sampling, we assign different adjective-noun pairs in each local region, ensuring that text embedding in one region do not affect the others.
> > >
> > > We would like to clarify that the primary focus of this paper is **composing multiple independently trained concept LoRA without conflicts**. The regionally-controllable sampling just serves as a means of multi-concept sampling to illustrate how our fused model excels at retaining each concept's identity. Therefore, **when comparing to other methods in Figure 6, we use the regionally-controllable sampling for all methods for fair comparison**. Our contribution mainly lies in our analysis how to reduce conflict in multi-concept customization and our solution: ED-LoRA+gradient fusion. And we validate our contribution not only by qualitative experiments, but also quantitative experiments. As in the attached PDF in the gloabl response, we compute CLIP-Image Alignment and conduct human preference study to validate our claim. We post the results for your reference. **Hope that our clarification can address your confusion regarding the primary contribution and focus of this paper.**
> > >
> > > ---
> > >
> > > **CLIP Image Alignment Evaluation**: We evaluate the CLIP image alignment of the sampling results **before and after concept fusion**. This evaluation allows us to quantify the identity loss that occurs during multi-concept fusion. As shown in the table below, our Mix-of-Show (ED-LoRA + Gradient Fusion) achieves the lowest identity loss during multi-concept fusion.
> > >
> > > | Methods | Real-Objects (Single→Fused) | Real-Characters (Single→Fused) | Real-Scenes (Single→Fused) | Mean Change |
> > > |  ---- | ----  | ----  | ----  | ----  |
> > > | LoRA + Weight Fusion | 0.864→0.778 (-0.086) | 0.761→0.555(-0.206) | 0.824→0.769 (-0.055) | 0.816→0.701 (-0.115) |
> > > | ED-LoRA + Weight Fusion |  0.868→0.798 (-0.070) | 0.802→0.634 (-0.168) | 0.858→0.816 (-0.042) | 0.843→0.749 (-0.094) |
> > > | ED-LoRA + Gradient Fusions | 0.868→0.846 (-0.022) | 0.802→0.770 (-0.032) | 0.858→0.838 (-0.020)  | **0.843→0.818 (-0.025)** |
> > > ---
> > >
> > > **Human Preference Study**: We utilize Amazon Mechanical Turk to conduct a human preference study on weight fusion and our gradient fusion for fusing ED-LoRAs. As demonstrated in the table below, users prefer our gradient fusion over weight fusion for multi-concept fusion.
> > >
> > > | Methods | Image Alignment | Text Alignment |
> > > |  ---- | ----  | ----  |
> > > | ED-LoRA + Weight Fusion |  33.5% | 47.5% |
> > > | ED-LoRA + Gradient Fusions | **66.5%** | **52.5%** |
> > >
> > > ---
> > >
> > >
> > > **Reference**
> > >
> > > [1] Training-Free Structure Diffusion Guidance For Compositional Text-to-Image Synthesis. ICLR2023

---

> > > > ### Comment · Reviewer_NQJc · 2023-08-19
> > > >
> > > > Thank the authors for the reply, and my concerns have been addressed. I will increase my score to weak accept.

---

> > > > > ### Author Response · Authors · 2023-08-20
> > > > > **Response to Reviewer NQJc**
> > > > >
> > > > > Dear Reviewer NQJc,
> > > > >
> > > > > We sincerely appreciate your time and effort in reviewing our paper, as well as your valuable and constructive feedback, thank you.
> > > > >
> > > > > Best regards,
> > > > >
> > > > > Authors of paper 6794.

---

### Official Review · Reviewer_TCkG · 2023-07-02

**Soundness:** 3 good
**Presentation:** 3 good
**Contribution:** 3 good
**Rating:** 6
**Confidence:** 2

**Summary:**

The paper presents Mix-of-Show, a framework that tackles the challenges of decentralized multi-concept customization in public large-scale text-to-image diffusion models. It introduces an embedding-decomposed LoRA (ED-LoRA) for single-client tuning and gradient fusion for the center node, enabling limitless concept fusion. The framework also incorporates regionally controllable sampling to address attribute binding and missing object issues. Experimental results demonstrate Mix-of-Show's ability to compose multiple customized concepts, including characters, objects, and scenes, with high fidelity. Overall, the framework offers a promising solution for decentralized multi-concept customization in text-to-image diffusion models.

**Strengths:**

The strength of Mix-of-Show lies in its comprehensive approach to decentralized multi-concept customization. It addresses the challenges of concept conflicts, identity loss, and missing objects through innovative techniques such as ED-LoRA, gradient fusion, and regionally controllable sampling. The framework enables the composition of complex and diverse concepts with high fidelity. It demonstrates promising results in generating multi-concept compositions, surpassing previous methods in terms of performance and flexibility.

**Weaknesses:**

The paper does not extensively discuss the computational complexity and resource requirements associated with the proposed framework.

While I'm not an expert in related area, the paper is generally well-written and easy to follow. At current stage, I have no further concerns regarding the technical details of the paper.

**Questions:**

Please refer to the above weakness.

**Limitations:**

Authors should demonstrate some failure cases in order to get a full grasp on the limitation of the proposed framework.

---

> ### Author Rebuttal · Authors · 2023-08-04
>
> Thank you for acknowledging our work. Part of the discussion on time complexity and limitations is included in Section 2.2 and Section 4.1 **in the supplementary material**, respectively. Here, we provide a more detailed discussion.
>
> **Notations**: W: Weakness, L: Limitation
>
> ---
>
> **W.1**: There are three parts in our methods, i.e., ED-LoRA, gradient fusion, and regionally-controllable sampling. We will separately discuss their time complexity and resource constraints.
>
> - ED-LoRA takes about 10 minutes for character concepts (about 15 images) and about 5 minutes for object concepts (about 5 images) on 2 A100 GPUs. With optimized techniques such as gradient checkpoint, xformer attention, and gradient accumulation provided by the open-source Diffusers repository [1], the training can fit into a single 8 GB GPU. The time complexity and memory cost are similar to LoRA [2]. It is worth noting that our methods do not require any regularization datasets like custom diffusion [3] and dreambooth [4], which helps reduce computation and data storage costs. For storing each concept, one ED-LoRA for each concept only takes 4.5 M storage, making it quite lightweight to share.
>
> - Gradient fusion requires a 12 GB memory GPU and about 24 and 50 minutes for composing 5 and 14 concepts on an A100 GPU, respectively. However, after further investigation, we found the bottleneck is in merging spatial layers (e.g., QKV in self-attention, Q in cross-attention) in Unet, where we should consider all sample timesteps. Our previous experiment was based on the DDIM scheduler, which required 50 timesteps. In our improved version, we switched to the more accurate DPM-Solver, which only takes 20 timesteps. Therefore, in this improved version, we only take 12 and 26 minutes to merge 5 and 14 concepts on an A100 GPU.
>
> - Regionally controllable sampling generates multiple concepts at once, and thus it requires nearly the same speed and memory cost to generate a 512 * 1024 image with T2I-Adapter, which takes around 4 seconds with a 10GB memory cost.
>
> ---
>
> **L.1**: We will discuss the limitations for ED-LoRA, gradient fusion, and regionally-controllable sampling, respectively.
>
> - In single concept tuning with ED-LoRA, it is challenging to generate full-body characters with detailed face texture in 512 * 512 resolution, as shown in Figure 1(b) in the attached PDF. This limitation is inherent to Stable Diffusion since the pretrained VAE compresses the information of the dedicated facial details. To overcome this limitation, when generating a full-body character, we need to adjust the aspect ratio (from 512 * 512 to 1024 * 512) to enlarge its facial region, as shown in the example in Figure 1 (b) in the attached PDF.
>
> - The limitation of gradient fusion lies in taking a relatively longer time compared to simple weight fusion. Even with our improvement discussed above, it still requires 12 and 26 minutes to merge 5 and 14 concepts. Further analysis on its redundancy to improve the time complexity will be left as future work.
>
> - In regionally controllable sampling, the limitation lies in the generated contents with structured backgrounds. For example, in Figure 7 of the supplementary material, the tree (left) and the railing (right) in the background lose quality and consistency when adopting regionally controllable sampling. This is because regionally controllable sampling breaks the interaction between regions, which should be further investigated to improve.
>
> ---
> **Reference**:
>
> [1] https://github.com/huggingface/diffusers
>
> [2] https://github.com/kohya-ss/sd-scripts
>
> [3] Multi-Concept Customization of Text-to-Image Diffusion. CVPR 2023.
>
> [4] DreamBooth: Fine Tuning Text-to-Image Diffusion Models for Subject-Driven Generation. CVPR 2023.

---

> ### Comment · Reviewer_TCkG · 2023-08-22
>
> I acknowledge that I have read authors' response as well as responses to other reviewers, authors have supplied the details of computation complexity regarding my concerns, and I'd like to maintain my rating.

---

### Official Review · Reviewer_acdR · 2023-07-05

**Soundness:** 3 good
**Presentation:** 3 good
**Contribution:** 2 fair
**Rating:** 6
**Confidence:** 3

**Summary:**

This paper focuses on the problem of multi-concept customization of diffusion model. Compared to the previous methods, this work focuses on fusing multiple customized objects into a single image with different extra conditions such as poses contours. The paper did an extensive study to analyze the roles of tuning the embedding and weights and based on this study it proposes a group of improvement to the multi-concepts customization. Based on the qualification results, the proposed methods are quite effective.

**Strengths:**

1. Compared to previous methods, this paper gives a more flexible concept customization framework where multiple concepts and different types of conditions are supported.
2. This paper provide some insights on how embedding and model weights affects the concept finetuning differently and propose effective  several modifications of the method accordingly.
3. Extensive qualification comparison is provided which proves the effectiveness of the proposed method.
4. The distributed training framework brough some insight on privacy preserving and federated learning related research.

**Weaknesses:**

This paper only provide qualification results with limited use cases. It would be better to provide some quantification analysis on more data, following some previous concept customization works.

**Questions:**

N/A

**Limitations:**

The author has address the limitations.

---

> ### Author Rebuttal · Authors · 2023-08-04
>
> Thanks for acknowledging our work. Regarding the weakness concerning quantitative experiments, we have included **extensive quantitative evaluation** in Section 3.1 of the supplementary material and added **quantitative analysis of our main components** (i.e., ED-LoRA, gradient fusion) in Table 1(a) in the attached PDF. Additionally, we conducted a **human preference study** on *Amazon Mechanical Turk* to further verify our gradient fusion, and the results are presented in Table 1(b, c) in the attached PDF. Here are the details about our quantitative experiments.
>
> 1) **Quantitative Evaluation**: In the supplementary material (Section 3.1.1 and Section 3.1.2), we have included our quantitative evaluation setting and results. Our evaluation is **spread across different methods** (i.e., LoRA [1], Custom Diffusion [2], and P+ [3]),  **covering all concept data**, including characters, objects, and scenes, and **benchmarking both the single-concept tuned and multi-concept fused model**. Through this extensive quantitative evaluation, we arrive at our conclusion: Mix-of-Show achieves similar single-concept tuning performance to LoRA but largely preserves the customized concept identity (measured by image-alignment) after multi-concept fusion.
>
> 2) **Quantitative Analysis of Main Components**: In the Table 1 of the attached PDF, we quantitative analyze the LoRA+weight fusion, ED-LoRA+weight fusion and ED-LoRA+gradient fusion across all concept data from different categories. From the results, we can find:
>
>    - With the weight fusion strategy, replacing vanilla LoRA with ED-LoRA improves image-alignment after multi-concept fusion.
>
>    - ED-LoRA+gradient fusion achieves the best image-alignment after multi-concept fusion.
>
>    These quantitative results strongly support our observation and methods.
>
> 3) **Human Preference Study**: In order to further verify the effectiveness of gradient fusion in multi-concept fusion, we conducted a human preference study on Amazon Mechanical Turk. We collected 400 questionnaires to assess human preference between gradient fusion and weight fusion for fusing our ED-LoRAs. The evaluation interface and results are summarized in Table 2(b, c) in the attached PDF. From the results, human raters significantly prefer gradient fusion over weight fusion (``66.5%`` vs. ``33.5%``) in fusing ED-LoRAs.
>
> Besides, we also provide more qualitative comparisons in Figure 3 and Figure 4 in the supplementary material, as well as Figure 1(c) in the attached PDF. Based on the above detailed qualitative and quantitative evaluations, we can see that Mix-of-Show **not only performs well in the limited use cases demonstrated in the main paper** but is also **generalizable across different data categories** (i.e., characters, objects, and scenes) and **different concept domains** (i.e., real-world concepts or anime concepts).
>
> ---
>
> **Reference**:
>
> [1] https://github.com/kohya-ss/sd-scripts
>
> [2] Multi-Concept Customization of Text-to-Image Diffusion. CVPR 2023.
>
> [3] P+: Extended Textual Conditioning in Text-to-Image Generation. Arxiv 2023.

---

> > ### Author Response · Authors · 2023-08-21
> > **Further Response To Reviewer acdR about Quantitative Analysis.**
> >
> > Dear Reviewer acdR,
> >
> > We have outlined our quantitative results in the rebuttal. The quantitative analysis of our main components, ED-LoRA and gradient fusion, is listed in Table 1 within the attached PDF of the global response. Additionally, the quantitative evaluation for comparison with previous methods (Custom Diffusion[1], LoRA[2], and P+[3]) is presented in both Table 1 and Table 2 in the supplementary materials.
> >
> > Here, we post the key results for your convinence. For the comprehensive quantitative evaluation, we kindly refer you to the submitted global response PDF and our supplementary materials. Hope those results can address your concerns.
> >
> > ---
> >
> > **1) Quantitative analysis of the effectiveness in reducing identity loss through ED-LoRA and Gradient Fusion.**
> >
> > **CLIP Image Alignment Evaluation**: We evaluate the CLIP image alignment of the sampling results **before and after concept fusion**. This evaluation allows us to quantify the identity loss that occurs during multi-concept fusion. As shown in the table below, our Mix-of-Show (ED-LoRA + Gradient Fusion) achieves the lowest identity loss during multi-concept fusion.
> >
> > | Methods | Real-Objects (Single→Fused) | Real-Characters (Single→Fused) | Real-Scenes (Single→Fused) | Mean Change |
> > |  ---- | ----  | ----  | ----  | ----  |
> > | LoRA + Weight Fusion | 0.864→0.778 (-0.086) | 0.761→0.555(-0.206) | 0.824→0.769 (-0.055) | 0.816→0.701 (-0.115) |
> > | ED-LoRA + Weight Fusion |  0.868→0.798 (-0.070) | 0.802→0.634 (-0.168) | 0.858→0.816 (-0.042) | 0.843→0.749 (-0.094) |
> > | ED-LoRA + Gradient Fusions | 0.868→0.846 (-0.022) | 0.802→0.770 (-0.032) | 0.858→0.838 (-0.020)  | **0.843→0.818 (-0.025)** |
> > ---
> >
> > **Human Preference Study**: We utilize Amazon Mechanical Turk to conduct a human preference study on weight fusion and our gradient fusion for fusing ED-LoRAs. As demonstrated in the table below, users prefer our gradient fusion over weight fusion for multi-concept fusion.
> >
> > | Methods | Image Alignment | Text Alignment |
> > |  ---- | ----  | ----  |
> > | ED-LoRA + Weight Fusion |  33.5% | 47.5% |
> > | ED-LoRA + Gradient Fusions | **66.5%** | **52.5%** |
> >
> > ---
> >
> > **2) Quantitative benchmark to previous methods.** We adopt CLIP image alignment following Custom Diffusion [1] to make a quantitative comparison with previous methods. In comparison to the embedding-only tuning approach (P+ [3]), our Mix-of-Show method achieves **much better image alignment**. When compared to Custom Diffusion [1] and LoRA [2], our method demonstrates **minimal identity loss**, resulting in **much better image alignment after multi-concept fusion**.
> >
> > | Methods | Real-Objects (Single→Fused) | Real-Characters (Single→Fused) | Real-Scenes (Single→Fused) | Mean Change |
> > |  ---- | ----  | ----  | ----  | ----  |
> > | P+ |  0.790→0.790 (-) | 0.670→0.670 (-) | 0.796→0.796 (-) | 0.752→0.752 (-) |
> > | Custom Diffusion | 0.842→0.808 (-0.034) | 0.714→0.694 (-0.020) | 0.804→0.750 (-0.054) | 0.787→0.751 (-0.036) |
> > | LoRA | 0.864→0.778 (-0.086) | 0.761→0.555(-0.206) | 0.824→0.769 (-0.055) | 0.816→0.701 (-0.115) |
> > | Mix-of-Show (Ours) | 0.868→0.846 (-0.022) | 0.802→0.770 (-0.032) | 0.858→0.838 (-0.020)  | **0.843→0.818 (-0.025)** |
> >
> > ---
> >
> > **Reference**:
> >
> > [1] Multi-Concept Customization of Text-to-Image Diffusion. CVPR 2023.
> >
> > [2] LoRA: https://github.com/kohya-ss/sd-scripts
> >
> > [3] P+: Extended Textual Conditioning in Text-to-Image Generation. Arxiv 2023.

---

### Official Review · Reviewer_Khyz · 2023-07-08

**Soundness:** 1 poor
**Presentation:** 1 poor
**Contribution:** 3 good
**Rating:** 4
**Confidence:** 4

**Summary:**

This paper addresses a problem of significant practical importance, which is how to employ a number of individually fine-tuned concept plug-ins to produce a model, which can then be used to generate an image that contains any combinations of the concepts. The key method is center node fusion, which trains a fused model instead of simple arithmetic averages over tuned Lora weights.

**Strengths:**

The paper tackles a problem of significant practical import, and it appears that the proposed method could work well. The center node fusion method is intuitive and can be implemented as is or tweaked to suit particular situations.

**Weaknesses:**

There are many. In no particular order,
- the term "decentralized" is quite misleading. What the paper meant was that the inverted embedding and weights for concept n must be trained by only knowing the data of concept n. That is not decentralized.
- observations from empirical observations do not add to the paper's technical contribution; in this reviewer's opinion, they are not convincing.
- fine details can influence the outcome a lot, for instance the split between the fine-tuning between embedding inversion (V is 1 token? 10 tokens?) and LoRA (dimensions).

**Questions:**

This reviewer does not have questions to the authors.

**Limitations:**

This work is neutral in terms of potential negative societal impact.

---

> ### Author Rebuttal · Authors · 2023-08-04
>
> Thanks for acknowledging the contribution of our work. We address the concerns as follows:
>
> **Notations**: W: Weakness
>
> ---
>
> **W.1**: Sorry. We might not fully grasp the meaning behind this statement, "What the paper meant was that the inverted embedding and weights for concept n must be trained by only knowing the data of concept n. That is not decentralized." **Are you suggesting that  decentralized learning is related to obtain individual concept models (embedding+weight) without utilizing data?**
>
> If it is the case, your understanding of the definition of decentralized learning could be incorrect. Decentralized learning does not imply that we cannot train the model of each client using data. Instead, it means the data from each client **cannot be shared** to obtain the fused model. For example, in the experiment setting in FedAVG [1], they split the CIFAR dataset into 100 clients, where each client hosts 600 samples. The data split of different clients will not be shared, but each client can train the model using their own data split.
>
> In the context of our multi-concept customization, we can use data to tune the ED-LoRA for each concept individually. The decentralization in our setting means that to obtain a multi-concept fused model, we do not need to access the source data of all concepts (1 to N). Instead, we can directly use gradient fusion to combine the ED-LoRA models of concepts (1 to N). This precisely captures the essence of decentralized learning.
>
> **If we have misunderstood your meaning in Weakness.1, please correct me during the discussion phase**.
>
> **W.2**: We respectfully disagree with this point. In academic research, empirical observations play a crucial role in motivating the development of reasonable methods and facilitating future research. For example, our Observation 1 about the role of embedding and weight in concept tuning motivated us to design ED-LoRA, and Observation 2 about identity loss in weight fusion motivated us to design gradient fusion. Without empirical observations, we could not have arrived at our final solution.
>
> However, to make our observations more convincing, we conducted **quantitative evaluations** to compare LoRA+weight fusion, ED-LoRA+weight fusion, and ED-LoRA+gradient fusion, as shown in Table 1(b, c) in the attached PDF. From the results, we can find:
>
> - With the weight fusion strategy, replacing vanilla LoRA with ED-LoRA improves image-alignment after multi-concept fusion.
> - ED-LoRA+gradient fusion achieves the best image-alignment after multi-concept fusion.
>
> These results also **strongly support** our observations and methods.
>
> **W.3**: The contribution of this work in the single-concept tuning part is to **unveil the distinct roles between embedding and weight, and their impact on concept fusion**. The conclusion is that we should transfer the in-domain essence of a given concept to the embedding and utilize the LoRA weight to capture out-domain information to avoid concept conflict, rather than relying solely on LoRA weight to encode all identity.
>
> The proposed ED-LoRA serves as a demonstration of a simple yet robust approach to achieve that goal. Using more tokens (e.g., 10) is probably not reasonable as it disrupts the grammar structure (adjective + noun) used to represent a concept, making it problematic to interact with other text prompts. The LoRA rank is indeed important, but we find that keeping a default low rank (r=4) is sufficient. A higher rank conflicts with our goal to shift in-domain essence to embedding. Consequently, the proposed ED-LoRA does not require extensive hyperparameter tuning for each concept, maintaining the same configuration for all experiments. Therefore, the performance does not come from changing any fine details.
>
> ---
> **Reference**:
>
> [1] Communication-Efficient Learning of Deep Networks from Decentralized Data. AISTATS 2017.

---

> > ### Comment · Reviewer_Khyz · 2023-08-21
> > **With regard to the term decentralized**
> >
> > In this reviewer's opinion, the term "decentralized" refers to having as little centralized control as possible, optionally compared to a commonly known centralized version.
> >
> > In the simple scenarios addressed in this submission, the Lora for concept n is trained entirely without any concern for the other nodes. I agree that this is de-centralized. However, this "train a lora for a concept" is also commonly done in a centralized training scenario, there is nothing that says a centralized node must mix the data of the n concepts to train one single Lora. So the contrast between centralized and de-centralized is very limited. Hence my remark that the use of the term is misleading.
> >
> > For the followoing, we can agree to disagree on the definition fo the term "decentralized", as I can be considered nitpicking.
> >
> > Federated learning, as when the term was coined, is a large number of distributed computations of gradients that are tightly controlled by a central node (parameter server). Furthermore, before launching the training, in practice a central node does careful leaf node qualification / balancing to ensure that good data enter federated training. As such, FedAvg is considered centralized training, not decentralized, by this reviewer.

---

> > > ### Author Response · Authors · 2023-08-21
> > > **Further Reply to Reviewer Khyz**
> > >
> > > Dear Reviewer Khyz,
> > >
> > > I agree that users can train separate concept ED-LoRA models and fuse them in the central node, as you mentioned in the context of a centralized setting. However, **our motivation lies in developing an algorithm suited for a real-world scenario** where users share tuned models rather than sharing their collected source data. For instance, these users might share ED-LoRA models based on Stable Diffusion on platforms such as Civitai [1] or Huggingface [2]. From our perspective, **this scenario is decentralized**. Nonetheless, our algorithm is also **applicable in a less-strict centralized setting as well**, owing to its merits compared to joint training: 1) Reusability and Scalability; 2) Mitigation of the feature mixing issue; 3) Reduction in difficulties when evaluating all concepts together.
> > >
> > > Although we retain some reservation regarding the precise definition of "decentralized," we will **provide a clearer explanation of this term in the revised paper**: although this algorithm is motivated by the decentralized scenarios; it also performs well in centralized settings.
> > >
> > > We appreciate your assistance in clarifying this concern.
> > >
> > > Best regards,
> > >
> > > Authors of Paper 6794.
> > >
> > > ---
> > >
> > > [1] https://civitai.com/
> > >
> > > [2] https://huggingface.co/

---

> ### Author Response · Authors · 2023-08-16
> **Response to Reviewer Khyz**
>
> Thank you for your valuable feedback on our submission. We have read your comments carefully and have addressed them in our rebuttal. As the second phase of the rebuttal process is ending soon, we would be grateful if you could acknowledge if our responses have addressed your comments. We would also be happy to engage in further discussions if needed. Thank you again for your time and consideration.

---

> ### Comment · Area_Chair_tAKU · 2023-08-20
>
> Dear Reviewer,
>
> Please check the new comments from the authors to see if they addressed your concerns.
>
> Regards,
> AC

---

### Official Review · Reviewer_B3ui · 2023-07-25

**Soundness:** 3 good
**Presentation:** 2 fair
**Contribution:** 2 fair
**Rating:** 5
**Confidence:** 4

**Summary:**

The paper proposes a framework called Mix-of-Show for decentralized multi-concept customization of diffusion models. The framework uses embedding-decomposed LoRA for single-client tuning and gradient fusion for center-node concept fusion to preserve the in-domain essence of single concepts and support theoretically limitless concept fusion. Additionally, it introduces regionally controllable sampling to extend spatially controllable sampling to address attribute binding and missing object problems in multi-concept sampling.

**Strengths:**

1. The paper proposes a framework called Mix-of-Show for decentralized multi-concept customization of diffusion models. It addresses the challenges of concept conflicts and identity loss during model fusion, and demonstrates the capability of composing multiple customized concepts with high fidelity.
2. The logical flow of the paper makes it easy for readers to understand and follow the research process.
3. The paper provides detailed analyses of the experimental results.

**Weaknesses:**

1. In Figure2(b), in the generated picture with text " V^{dogA}, a V^{cat} and a V^{dogB}, in a living room.", is the text wrong here? Or is this picture exactly the output of the model? I'm a little confused because the background of the picture is suburb rather than living room.
2. In line 230 to 231, you mentioned that Figure3(f) demonstrates that the gradient fusion preserves the concept identity, but I think the effect of this is not obvious. In other words, in the method that only uses ED-LoRA with weight fusion, the problem of identity loss basically does not exist. I can't see much difference between these two pictures. Maybe Thanos in Figure7(c) is a better example. It's a good way to find some special examples to show the effect, but I'd like to know if there is a quantitative way to accurately evaluate the effectiveness of this fusion approach.
3. In section 4, you made a qualitative comparison. Have you thought about using a quantitative approach such as CLIP score to compare your model with others, or have you ever thought about developing an new evaluation metric which can accurately tests your own approach?
4. When you use the method you proposed, have you encountered any poor-quality images? Have you done an error analysis? And have you analysed the limitations and shortcomings of these methods?
5. Some equations lacks essential explanatory notes for some of the mathematical symbols used:
Equation 1, the notation of $\epsilon$ is missing
6. Some figure misses detailed explanation which causes confusion (See question 2 and 3)
7. Lack of details concerning the collected dataset and the text encoder used in the study.
8. The dataset contains different characters, objects, and scenes, however, I only see the demonstration of characters. More examples about different objects and scenes can be provided to make the results more solid.
9. There is a lack of quantitative analysis in this work. The effect of the method proposed in this paper is represented by the sample pictures, which is a qualitative analysis and can not well reflect the generalizability of the method.

**Questions:**

Question:
1. How do you initialize V+_{class} in Line 194? Can you provide a more detailed explanation?
2. In Figure 2, what does @ mean?
3. In Figure 2, what is - [x1, y1, x2, y2]? Are they the spatial coordinates of different concepts?
4. Did you consider conducting human evaluation to make the results more convincing?

Suggestions:
1. The related work content of the paper is less. It would be beneficial to expand a more detailed explanation of the relevant content.
2. A visual explanation of the overall architecture that includes the individual modules is useful, as it will help the readers to better understand the structure of the new framework.
3. The experiment section needs to introduce more details about the dataset, such as data quantity, source, input. This would provide readers with a better understanding of experiment details.

**Limitations:**

Yes

---

> ### Author Rebuttal · Authors · 2023-08-05
>
> Thank you for providing many detailed comments. We will address your concerns by referring to the relevant contents in the supplementary material and the attached PDF (in the global response).
>
> **Notations**: W: Weakness, Q: Question, S: Suggestion
>
> ---
>
> **W.1**: Thank you for pointing that out. It was a typo. The correct text prompt should be "a $V^{dogA}$, a $V^{cat}$, and a $V^{dogB}$, on the grass, under the sunset." We will correct it in the revised version.
>
> **W.2**: Thanks for raising this concern. We admit that it might be challenging for readers who are not familiar with those anime characters to see the difference. To facilitate a better comparison between weight fusion and gradient fusion for fusing our ED-LoRAs, we provide **more qualitative examples of real-world characters and objects** in Figure 1(a) of the attached PDF. Additionally, we conduct **two quantitative evaluations** to compare weight fusion and gradient fusion:
>
> - We evaluate the **CLIP image alignment** before and after multi-concept fusion to measure the subject identity loss, in the Table 1(a) of the attached PDF. From the result, we can find gradient fusion achieves less image-alignment loss compared to weight fusion for fusing ED-LoRAs.
>
> - We perform a **human preference study** on Amazon Mechanical Turk. We collected 400 questionnaires to assess human preference for these two fusion methods. The evaluation interface and results are summarized in Table 2(b, c) in the attached PDF. Based on the results, human raters believe that gradient fusion better preserves subject identity compared to weight fusion (`66.5%` vs. `33.5%`) when fusing ED-LoRAs.
>
> **W.3**: We have conducted a quantitative evaluation following Custom Diffusion [1], which involves utilizing the CLIP text model and CLIP image model to assess text alignment and image alignment. For detailed settings and results of our quantitative evaluation, please refer to Section 3.1.1 and Section 3.1.2 in the supplementary material.
>
> **W.4**: Limitations of gradient fusion and regionally controllable sampling have been discussed in Section 4.1 of the supplementary material. Additionally, the limitation of ED-LoRA lies in the inherent bottleneck of Stable-Diffusion, resulting in poor-quality full-body character generation, as shown in Figure 1(b) in the attached PDF. However, this limitation can be overcome by adjusting the aspect ratio to increase the size of the facial region.
>
> **W.5**: Thank you for pointing that out. We will thoroughly proofread the paper and ensure that the missing explanatory notes are included. In Equation 1, the symbol $\epsilon$ represents the noised feature map at timestep $t$, which serves as the ground truth for the denoiser UNet prediction.
>
> **W.6**: Thank you for pointing that out. We will revise the related figure captions in this paper to add detailed instructions to guide the comparison of our results.
>
> **W.7**: The description of evaluation datasets and data quantity of each concept are included in Section 2.1 and Table 2 of the supplementary, and the representative image of each concept is shown in Figure 1 and Figure 2 in the main paper. Our experiment is based on the Stable Diffusion, which utilizes the pretrained CLIP text encoder.
>
> **W.8**: Thank you for the suggestions. Given that characters are **the most challenging** in concept customization due to their intricate facial details, most of our results are showcased using characters. We also include more examples for objects and scenes in the Figure 4 of the supplementary and Figure 1(c) of the attached PDF.
>
> **W.9**: Thanks for pointing that out. We include the quantitative analysis in Table 1(a) of the attached PDF, comparing LoRA+weight fusion, ED-LoRA+weight fusion, and ED-LoRA+gradient fusion. From the results, we clearly find:
> - With the weight fusion strategy, replacing vanilla LoRA with ED-LoRA reduces the image-alignment loss after multi-concept fusion.
> - ED-LoRA+gradient fusion achieves **the least image-alignment loss** after multi-concept fusion.
>
> Additionally, we conducted a human preference study, as discussed in **W.2**, to verify the effectiveness of gradient fusion.
>
> These quantitative analyses **strongly support our observations and methods**. We will add these quantitative analyses to the main paper in the revised version.
>
> ---
>
> **Q.1**: The $V^{+}_{class}$ is initialized by the class token, representing the super-class of the given concept. For example, for the concept "dogA," it is initialized by "dog," and for "Hermione," it is initialized by "woman."
>
> **Q.2-Q.3**: Do you mean Figure 5 (not Figure 2)? In Figure 5, the "@" symbol represents matrix multiplication. The values [x1, y1, x2, y2] correspond to the spatial coordinates of each concept. In the revised version, we will add an illustration of these notations in the caption.
>
> **Q.4**: Sure, as discussed in **W.2**, we conduct a **human preference study** on Amazon Mechanical Turk to compare the gradient fusion and weight fusion for fusing our ED-LoRAs. From the results, human raters believe that gradient fusion better preserves subject identity compared to weight fusion (`66.5%` vs. `33.5%`).
>
> ---
>
> **S.1**: Thanks for the suggestion. Due to the page limit, we only discuss the tuning-based customization works, which most related to this work. In the revised version, we will expand the discussion of existing related works and also add the discussion of tuning-free customization works.
>
> **S.2**: Thanks for the suggestion. In the revised version, we will further expand the details of the framework shown in the main paper (Figure 4).
>
> **S.3**: Please refer to **W.7** and Section 2.1 in the supplementary material for our dataset and implementation details. We will move some important contents into the main paper in the revised version.
>
> ---
> **Reference**:
>
> [1] Multi-concept customization of text-to-image diffusion. CVPR 2023.

---

> ### Author Response · Authors · 2023-08-16
> **Response to Reviewer B3ui**
>
> Thank you for your valuable feedback on our submission. We have read your comments carefully and have addressed them in our rebuttal. As the second phase of the rebuttal process is ending soon, we would be grateful if you could acknowledge if our responses have addressed your comments. We would also be happy to engage in further discussions if needed. Thank you again for your time and consideration.

---

> ### Comment · Reviewer_B3ui · 2023-08-16
> **Comments**
>
> I will keep my score unchanged as no further evaluation has been provided to convince me otherwise.

---

> > ### Author Response · Authors · 2023-08-16
> > **Further feedback about evaluation results.**
> >
> > Dear Reviewer B3ui,
> >
> > **We have included the quantitative evaluation and analysis of our Mix-of-Show in the supplementary materials and the attached PDF in our global response.** The quantitative analysis of our main components, ED-LoRA and gradient fusion, is listed in Table 1 within the attached PDF of the global response. Additionally, the quantitative evaluation for comparison with previous methods (Custom Diffusion[1], LoRA[2], and P+[3]) is presented in both Table 1 and Table 2 in the supplementary materials.
> >
> > For your convenience and reference, we list a portion of the important results as follows. For the comprehensive quantitative evaluation, we kindly refer you to the submitted global response PDF and our supplementary materials.
> >
> > ---
> >
> > **1) Quantitative analysis of the effectiveness in reducing identity loss through ED-LoRA and Gradient Fusion.**
> >
> > **CLIP Image Alignment Evaluation**: We evaluate the CLIP image alignment of the sampling results **before and after concept fusion**. This evaluation allows us to quantify the identity loss that occurs during multi-concept fusion. As shown in the table below, our Mix-of-Show (ED-LoRA + Gradient Fusion) achieves the lowest identity loss during multi-concept fusion.
> >
> > | Methods | Real-Objects (Single→Fused) | Real-Characters (Single→Fused) | Real-Scenes (Single→Fused) | Mean Change |
> > |  ---- | ----  | ----  | ----  | ----  |
> > | LoRA + Weight Fusion | 0.864→0.778 (-0.086) | 0.761→0.555(-0.206) | 0.824→0.769 (-0.055) | 0.816→0.701 (-0.115) |
> > | ED-LoRA + Weight Fusion |  0.868→0.798 (-0.070) | 0.802→0.634 (-0.168) | 0.858→0.816 (-0.042) | 0.843→0.749 (-0.094) |
> > | ED-LoRA + Gradient Fusions | 0.868→0.846 (-0.022) | 0.802→0.770 (-0.032) | 0.858→0.838 (-0.020)  | **0.843→0.818 (-0.025)** |
> > ---
> >
> > **Human Preference Study**: We utilize Amazon Mechanical Turk to conduct a human preference study on weight fusion and our gradient fusion for fusing ED-LoRAs. As demonstrated in the table below, users prefer our gradient fusion over weight fusion for multi-concept fusion.
> >
> > | Methods | Image Alignment | Text Alignment |
> > |  ---- | ----  | ----  |
> > | ED-LoRA + Weight Fusion |  33.5% | 47.5% |
> > | ED-LoRA + Gradient Fusions | **66.5%** | **52.5%** |
> >
> > ---
> >
> > **2) Quantitative benchmark to previous methods.** We adopt CLIP image alignment following Custom Diffusion [1] to make a quantitative comparison with previous methods. In comparison to the embedding-only tuning approach (P+ [3]), our Mix-of-Show method achieves **much better image alignment**. When compared to Custom Diffusion [1] and LoRA [2], our method demonstrates **minimal identity loss**, resulting in **much better image alignment after multi-concept fusion**.
> >
> > | Methods | Real-Objects (Single→Fused) | Real-Characters (Single→Fused) | Real-Scenes (Single→Fused) | Mean Change |
> > |  ---- | ----  | ----  | ----  | ----  |
> > | P+ |  0.790→0.790 (-) | 0.670→0.670 (-) | 0.796→0.796 (-) | 0.752→0.752 (-) |
> > | Custom Diffusion | 0.842→0.808 (-0.034) | 0.714→0.694 (-0.020) | 0.804→0.750 (-0.054) | 0.787→0.751 (-0.036) |
> > | LoRA | 0.864→0.778 (-0.086) | 0.761→0.555(-0.206) | 0.824→0.769 (-0.055) | 0.816→0.701 (-0.115) |
> > | Mix-of-Show (Ours) | 0.868→0.846 (-0.022) | 0.802→0.770 (-0.032) | 0.858→0.838 (-0.020)  | **0.843→0.818 (-0.025)** |
> >
> > ---
> >
> > **Reference**:
> >
> > [1] Multi-Concept Customization of Text-to-Image Diffusion. CVPR 2023.
> >
> > [2] LoRA: https://github.com/kohya-ss/sd-scripts
> >
> > [3] P+: Extended Textual Conditioning in Text-to-Image Generation. Arxiv 2023.

---

> > > ### Author Response · Authors · 2023-08-19
> > > **To Reviewer B3ui**
> > >
> > > Dear Reviewer B3ui,
> > >
> > > Thank you once again for providing constructive review on our paper.
> > >
> > > With the discussion period ending in two days, we wish to emphasize that we **have conducted a comprehensive quantitative evaluation to validate our primary contribution**, ED-LoRA and gradient fusion. Furthermore, we have **included a quantitative comparison with previous methods**. We have attached these results above for your convenience.
> > >
> > > We hope these quantitative evaluations will address your concerns. We would appreciate it if you could review those results and provide feedback for us.
> > >
> > >
> > > Best regards,
> > >
> > > Authors of Paper 6794.

---

> > ### Comment · Reviewer_B3ui · 2023-08-22
> > **About the new results**
> >
> > The new results have convinced me to some extent, but compared to Custom Diffusion and others, the improvement is not significant. However, considering the novelty of this method, I have raised my score.

---

> ### Comment · Area_Chair_tAKU · 2023-08-20
>
> Dear Reviewer B3ui,
>
> Please check the new comments from the authors to see if they addressed your concerns.
>
> Regards,
> AC

---

### Author Rebuttal · Authors · 2023-08-07

Thank you to all the reviewers for providing valuable comments to help us improve our submission. We would like to clarify that, due to the page limit, the **quantitative evaluation**, **dataset details**, **limitation discussion** and **potential negative society impact discussion** are provided **in the supplementary material**. In the following response, we will refer you to the relevant section. We plan to move parts of important quantitative results and discussions to the main paper in the revised version. Additionally, we have included **supporting figures and tables in the attached PDF for this response**.

---

### Decision · Program_Chairs · 2023-09-21

**Decision:**

Accept (poster)

**Comment:**

This paper received a range of scores, with four reviewers recommending acceptance and one reviewer suggesting rejection. The authors effectively addressed the reviewers' comments during the rebuttal process. The paper introduces a novel framework called "Mix-of-Show" to tackle the challenge of decentralized multi-concept customization of diffusion models. The method shows promising results and outperforms existing approaches in the domain of multi-concept customization. However, it is suggested that more practical results should be included. In light of the overall assessment and improvements made by the authors, the AC has decided to accept this paper.